# Feature Enhancement Method of Rolling Bearing Based on K-Adaptive VMD and RBF-Fuzzy Entropy

**DOI:** 10.3390/e24020197

**Published:** 2022-01-27

**Authors:** Jing Jiao, Jianhai Yue, Di Pei

**Affiliations:** School of Mechanical, Electronic and Control Engineering, Beijing Jiaotong University, Beijng 100044, China; jingjiao@bjtu.edu.cn (J.J.); pei_di@bjtu.edu.cn (D.P.)

**Keywords:** variational mode decomposition, fuzzy entropy, feature enhancing, rolling element bearing

## Abstract

The complex and harsh working environment of rolling bearings cause the fault characteristics in vibration signal contaminated by the noise, which make fault diagnosis difficult. In this paper, a feature enhancement method of rolling bearing signal based on variational mode decomposition with *K* determined adaptively (K-adaptive VMD), and radial based function fuzzy entropy (RBF-FuzzyEn), is proposed. Firstly, a phenomenon called abnormal decline of center frequency (ADCF) is defined in order to determine the parameter *K* of VMD adaptively. Then, the raw signal is separated into *K* intrinsic mode functions (IMFs). A coefficient *En* for selecting optimal IMFs is calculated based on the center frequency bands (CFBs) of all IMFs and frequency spectrum for original signal autocorrelation operation. After that, the optimal IMFs of which *En* are bigger than the threshold are selected to reconstruct signal. Secondly, RBF is introduced as an innovative fuzzy function to enhance the feature discrimination of fuzzy entropy between bearings in different states. A specific way for determination of parameter *r* in fuzzy function is also presented. Finally, RBF-FuzzyEn is used to extract features of reconstructed signal. Simulation and experiment results show that K-adaptive VMD can effectively reduce the noise and enhance the fault characteristics; RBF-FuzzyEn has strong feature differentiation, superior noise robustness, and low dependence on data length.

## 1. Introduction

Rolling bearing is one of the most widely used components in mechanical equipment. The working environment of bearing is relatively bad, and the failure rate is high [1]. Once a failure of bearing occurs, it will seriously affect the safe operation of equipment. Thus, measurement for rolling bearings plays an increasingly important role. Adamczak et al. [2] compared various measurement systems for evaluating the vibrations of rolling bearings, which can reflect the statement of bearing. At present, fault diagnosis of rolling bearings mainly focuses on signal noise reduction and feature extraction. The commonly used denoising methods for rolling bearing vibration signal mainly include wavelet analysis [3], empirical mode decomposition (EMD) [4], ensemble empirical mode decomposition (EEMD) [5], local mean decomposition (LMD) [6], singular value decomposition (SVD) [7], stochastic resonance [8], and so on.

In 2014, Dragomiretskiy and Zosso proposed a new signal mode decomposition method, namely variational mode decomposition (VMD) [9]. VMD employs the variational frame and alternating direction multiplier method to decompose the mode of signal and extract the center frequency of each mode at the same time. It overcomes the shortcoming of mode aliasing in EMD. The essence of VMD is multiple adaptive Wiener filter banks, which has a solid theoretical foundation. The advantages of VMD make it widely used in the field of rolling bearing fault diagnosis. However, there remain two main problems to be solved in realizing noise reduction of rolling bearing vibration data by VMD: determine the number of modes *K* and penalty factor *α*; select IMFs and reconstruct the signal. Scholars have made some research achievements on these two issues.

The existing researches of rolling bearing fault diagnosis based on VMD mainly consist of two kinds of technology roadmaps for the solution of problem one. The one with more abundant research determines an objective function or fitness function, and gets optimal value of function by iterating different *K* or [*K*, *α*]. For example, Li et al. [10] determined *K* based on the kurtosis of Hilbert envelope spectrum. Dibaj et al. [11] acquired multiple groups of [*K*, *α*] according to the correlation function of adjacent components, and optimized parameters of VMD so that the decomposed modes have minimum bandwidth. Gong et al. [12] applied the component kurtosis difference between two adjacent *K* values as the target value to get *K*. Li et al. [13] determined optimal *K* according to the proposed energy loss coefficient *e* and an upper limit 14 of *K*. The second type for determining parameters of VMD is based on observing the characteristics of signal waveform. Chen et al. [14] proposed a warped variable mode decomposition (WVMD) applying to variable speed signal decomposition of rotating machines, taking the short time Fourier transform (STFT) of signal as time-frequency representations (TFR), and determining the number of modes *K* by discovering characteristics of TFR. Wu et al. [15] developed a waveform method to determine the mode number by observing whether multiple frequencies with remarkable amplitudes occur in frequency spectrum of each component when *K* takes different values. Qi et al. [16] performed STFT on original signal, drew a time-frequency diagram, and estimated the number of modes *K* based on the diagram. In addition to the above two categories, some scholars optimize VMD parameters through various methods. Jiang et al. [17] introduced a method to estimate initial center frequency (ICF) based on energy fluctuation spectrum (EFS), and optimized VMD parameters according to the change of ICF. Li et al. [18] proposed VMD fractional Fourier transform (VMD-FRFT), which is insensitive to the decomposition layers *K*, so that the decomposition result will not affected by parameter *K*. Aiming at problem two, many scholars make use of time domain feature, frequency domain feature, time-frequency domain feature, entropy, etc., or propose improved indicators to select IMF. Papers [12,13,19] took kurtosis as an index, or combine kurtosis with other coefficient and methods to achieve component screening. Paper [11] defined a weighted kurtosis index (WKI), and the IMF corresponding to the largest WKI is selected. Paper [10] determined the optimal IMF through band entropy. In addition, some scholars put forward indicators representing the relationship between IMF as a basis for selection. A mutual information index was defined based on the entropy of data to distinguish effective modes from noise modes in paper [18].

On the abovementioned works for bearing faults based on VMD, a range or an upper limit of *K* is set in most of the solutions for problem one. The maximum *K* is generally limited to about 15. However, setting an upper limit for *K* may result in under decomposition of signal. We found that with the increase of *K*, frequency distribution of IMF becomes more concentrated, and signal is split more finely in the whole frequency band based on a great number of simulation and experimental signal analysis. Thus, the increase of *K* is conducive to separate signal and noise more accurately and reduce noise content in optimal component. In this paper, we proposed K-adaptive VMD, which does not set an upper limit for *K*, and adaptively determines *K* according to the abnormal decline of center frequency disorder (ADCF) of IMFs. Aiming at problem two, most of the researches are based on statistical indicators of signals in time or frequency domain, such as kurtosis, permutation entropy, envelope spectral kurtosis, correlation coefficient, Euclidean distance and mutual information, etc. These indicators judge the distribution of components in the whole time or frequency domain, which is more vulnerable to noise interference. A method for IMF selecting based on spectrum energy *En* of autocorrelation operation is proposed in this paper to solve this problem. CFB is determined according to the distribution of IMF spectrum, so that the energy of frequency spectrum for IMF on CFB accounts for *b* times of the whole frequency spectrum. Taking the frequency spectrum of autocorrelation operation for original signal as analysis object, energy of the analysis object on CFB is computed. Thus, the calculation method for *En* limits the range to CFB.

The signal-to-noise ratio (SNR) will be greatly improved, and the fault characteristics in the signal will be more obvious after decomposition and reconstruction by K-adaptive VMD. However, fault diagnosis based on spectrum analysis still requires a lot of empirical knowledge. Thus, it is necessary to extract the feature of reconstructed signal. The commonly used characteristics of vibration signals include time domain, frequency domain and entropy features. Among these features, entropy has been widely employed because it can preferably represent statistical characteristics, dynamic change and complexity of signal [20].

Approximate entropy, sample entropy and fuzzy entropy have been applied more widely among all entropy features, because they can measure the complexity of data effectively. In 1991, Pincus [21] proposed approximate entropy to evaluate the complexity of short data sequences. Although approximate entropy has good anti-interference ability, it is easily biased caused by self-matching [22]. Richman et al. [22] developed sample entropy in 2000 based on this problem. Sample entropy is closer to statistical theory than approximate entropy, and has low dependence on data length. However, affected by the similarity definition, approximate entropy and sample entropy hold poor consistency. In order to solve this problem, Chen et al. [23] carried out fuzzy entropy in 2007, which has the advantages of robustness to background noise, strong anti-interference ability but still depend on data length. Fuzzy entropy, approximate entropy and sample entropy all need to split the input data, work out distance matrix of coarse sequence, and reckon the similarity of sequence according to distance matrix. The essential difference between these three entropy lies in the method of calculating similarity.

Approximate entropy and sample entropy take the Heaviside function (unit step function) of distance as similarity, while fuzzy entropy estimates the similarity through the exponential function of distance, which is called fuzzy function. In 2009, Chen et al. [24] compared fuzzy entropy with approximate entropy and sample entropy. It is concluded that fuzzy entropy is superior to the other two entropies in expressing signal complexity, continuity and relative consistency with changes in its own parameters, dependence on data length, noise robustness and so on. In addition, the fuzzy function has specific limitations. Functions that meet these limitations can be operated as fuzzy function. Moreover, the limitations of fuzzy function are easy to meet, which provides more probability for the application and improvement of fuzzy entropy. However, all of the existing fuzzy functions contain two parameters, *n* and *r*, and there is no certain way to determine the parameters. Radial based function (RBF) is introduced as the fuzzy function to solve this problem. The function contains only one parameter *r*, and the detail determination method and derivation process of *r* is proposed.

The specific implementation steps of the rolling bearing vibration signal feature enhancement method based on K-adaptive VMD and RBF-FuzzyEn proposed in this paper are listed hereafter. Firstly, based on the change in center frequency of IMFs, the *K* before ADCF appears twice in succession is utilized as the number of decomposed modes to get the largest *K* as possible. Then, center frequency bands (CFBs) of all components are computed, and energy *En* of frequency spectrum for original signal autocorrelation operation on CFBs are obtained. After that, select optimal components which *En* are greater than threshold, and reconstruct the de-noised signal. Finally, parameter *r* in fuzzy function is determined according to denoising signal, and the feature of signal is extracted by RBF-FuzzyEn.

This paper is organized as followings. In Section 2, we introduce the calculation method of VMD and fuzzy entropy. The specific implementation process of K-adaptive VMD and RBF-FuzzyEn proposed in this paper are introduced in Section 3. Section 4 shows simulation analysis and verification based on non-linear amplitude modulation and frequency modulation (AM-FM) simulation signal and the rolling bearing fault simulation signal. Section 5 gives the application of the proposed method in rolling bearing feature enhancement. Conclusions are enclosed in Section 6.

## 2. Basic Theory

### 2.1. VMD

VMD decomposes the input signal *x* into *K* modes *u_k_*(*t*) with a certain bandwidth and center frequency, so as to minimize the sum of bandwidths for each mode [9]. In order to reckon the bandwidth of each mode, Hilbert transform is carried out on *u_k_*(*t*), and its unilateral spectrum is obtained. Then the unilateral spectrum is mixed with an estimated center frequency and integrated into the baseband. Finally, the *L*_2_ norm square is calculated for the gradient of fundamental band mode, and the following variational constraint problem is worked out:(1)$$\{\begin{array}{c} \underset{\left\{u_{k}\right\},\left\{\omega_{k}\right\}}{min}\left\{\overset{K}{\underset{k=1}{\sum }}∥\partial_{t}\left[\left(\delta \left(t\right)+\frac{j}{\pi t}\right)*u_{k}\left(t\right)\right]e^{-j\omega_{k}t}{∥}_{2}^{2}\right\}\\ s.t.\overset{K}{\underset{k=1}{\sum }}u_{k}=x \end{array}$$
where *K* is the total number of modes; *u_k_* is the decomposed single-component AM-FM signal; *ω* is the center frequency of each component; *x* is original signal.

The quadratic penalty factor *α* and Lagrangian multiplication operator *λ* are introduced in order to transform the above constrained problem into an unconstrained optimization problem. The augmented Lagrangian function is as follows:(2)$$L\left(\left\{u_{k}\right\},\left\{\omega_{k}\right\},\lambda \right)=\alpha \overset{K}{\underset{k=1}{\sum }}∥\partial_{t}\left[\left(\delta \left(t\right)+\frac{j}{\pi t}\right)*u_{k}\left(t\right)\right]e^{-j\omega_{k}t}{∥}_{2}^{2}+∥x\left(t\right)-\overset{K}{\underset{k=1}{\sum }}u_{k}\left(t\right){∥}_{2}^{2}+\lambda \left(t\right),x\left(t\right)-\overset{K}{\underset{k=1}{\sum }}u_{k}\left(t\right)$$

Equation (2) can be equivalently decomposed into sub-problems of alternately finding *u_k_* and *ω_k_* through Alternate Direction Method of Multipliers (ADMM). The calculation formulas of ${\hat{u}}_{k}^{n+1}$ and $\omega_{k}^{n+1}$ are shown in Equations (3) and (4).
(3)$${\hat{u}}_{k}^{n+1}\left(\omega \right)=\frac{\hat{x}\left(\omega \right)-\sum_{i<k}{\hat{u}}_{i}^{n+1}\left(\omega \right)-\sum_{i>k}{\hat{u}}_{i}^{n}\left(\omega \right)+\frac{{\hat{\lambda }}^{n}\left(\omega \right)}{2}}{1+2\alpha {\left(\omega -\omega_{k}^{n}\right)}^{2}}$$
(4)$$\omega_{k}^{n+1}=\frac{\int_{0}^{\infty }\omega {\left|{\hat{u}}_{k}^{n+1}\left(\omega \right)\right|}^{2}d\omega }{\int_{0}^{\infty }{\left|{\hat{u}}_{k}^{n+1}\left(\omega \right)\right|}^{2}d\omega }$$
where, $\hat{x}\left(\omega \right)$, $\hat{u}\left(\omega \right)$ and $\hat{\lambda }\left(\omega \right)$ are the Fourier transform of x(t), u(t) and λ(t), respectively. After $\hat{u}\left(\omega \right)$ the updating of $\omega_{k}$ and $\hat{\lambda }$ should be calculated by the following formula:
(5)$${\hat{\lambda }}^{n+1}\left(\omega \right)\;=\;{\hat{\lambda }}^{n}\left(\omega \right)+\tau \left(\hat{x}\left(\omega \right)-\underset{k}{\sum }{\hat{u}}_{k}^{n+1}\left(\omega \right)\right)$$
where *τ* denotes the tolerance of noise. The optimization will end until the following condition is satisfied.
(6)$$\sum_{k}\frac{∥{\hat{u}}_{k}^{n+1}-{\hat{u}}_{k}^{n}{∥}_{2}^{2}}{{\hat{u}}_{k}^{n}{}_{2}^{2}}<\varepsilon$$
where *ε* is the tolerance of convergence criterion.

### 2.2. Fuzzy Entropy

Fuzzy entropy, proposed by Chen et al. [23], is a widely used statistical feature, which is applied to measure the complexity of data. The specific calculation steps are listed as below:
**Step 1.** Get the coarse-grained data. Work out X(i) from the original signal sequence *x* based on Equation (7).
(7)$$\{\begin{array}{c} X\left(i\right)\;=\;\left[x\left(i\right),x\left(i+1\right),\ldots ,x\left(i+m-1\right)\right]-x_{0}\left(i\right)\;i\;=\;1,\ldots ,\;N-m+1\;\\ x_{0}\left(i\right)\;=\;\frac{1}{m}\overset{m-1}{\underset{l\;=\;0}{\sum }}x\left(i+l\right) \end{array}$$
where *N* is the length of *x*; *m* is the length of coarse-grained sequence *X*(*i*). Referring to paper [23], the value of *m* is taken as 2.

**Step 2.** Calculate the maximum absolute distance $d_{ij}^{m}$ between *X*(*i*) and *X*(*j*).
(8)$$d_{ij}^{m}=d\left[X\left(i\right),X\left(j\right)\right]=\underset{k\in \left(0,m-1\right)}{\max}\left|\left(x\left(i+k\right)-x_{0}\left(i\right)\right)-\left(x\left(j+k\right)-x_{0}\left(j\right)\right)\right|\;j\neq i$$

**Step 3.** Work out similarity $D_{ij}^{m}$ between *X*(*i*) and *X*(*j*) through fuzzy function based on maximum absolute distance $d_{ij}^{m}$.
(9)$$D_{ij}^{m}=f\left(d_{ij}^{m}\right)$$

The fuzzy function in paper [23] is $f_{1}\left(d_{ij}^{m}\right)\;=\;\exp\left(-{\left(d_{ij}^{m}\right)}^{n}/r\right)$, and in [24] is $f_{2}\left(d_{ij}^{m}\right)=\exp\left(-{\left(d_{ij}^{m}/r\right)}^{n}\right)$. Both functions contain two parameters *r* and *n*, which determine the width and the gradient of boundary for the fuzzy functions respectively. The forms of fuzzy functions are diverse, but must meet the following limitations: the function possesses continuity; the function should be convex, which maximizes the self-similarity.

**Step 4.** Take *m* = *m* + 1, and repeat steps 2 and 3.

**Step 5.** Calculate fuzzy entropy based on Equations (10) and (11).


(10)
$$\mathrm{FuzzyEn}\left(m,n,r,N\right)=\ln\varphi^{m}\left(n,r\right)-\ln\varphi^{m+1}\left(n,r\right)$$



(11)
$$\varphi^{m}\left(n,r\right)=\frac{1}{N-m}\overset{N-m}{\underset{i=1}{\sum }}\left(\frac{1}{N-m-1}\overset{N-m}{\underset{j=1,j\neq i}{\sum }}D_{ij}^{m}\right)$$


## 3. K-Adaptive VMD and RBF-FuzzyEn

### 3.1. K-Adaptive VMD

We proposed a signal decomposition method K-adaptive VMD, which does not set an upper limit of *K*. Based on whether there is ADCF in IMF, the maximum number of mode *K* is obtained. The calculation process of K-adaptive VMD is shown in Figure 1, which is divided into two parts: *K* value determination and component selecting.

#### 3.1.1. Determination of *K*

Parameters *K* and *α* in VMD have certain impact on decomposition results, but many scholars have pointed out that the key to VMD lies in the optimization of decomposition layers *K* [12,18,25]. The determination of *K* is an extremely important step in VMD. When α is fixed, for small *K* value, multiple components of the original signal may appear in one IMF simultaneously, cause the signal and noise cannot be separated effectively, which is called under decomposition; for big *K* value, some components of the original signal will be decomposed into multiple IMFs, which is called over decomposition. But the problem of over-decomposition can be solved by selecting and reconstructing the corresponding components. With the increase of *K*, the signal is separated more finely in the whole frequency band, and the energy of the component is more concentrated. Through a great number of simulation and experimental signal research, we found that the center frequency of IMFs will decrease at some components, which is called ADCF. In addition to *K* and *α*, VMD also contains two parameters: *τ* and *ε*, which represent the tolerance of noise and the tolerance of convergence criterion, respectively. *τ* affects the Lagrange multiplier *λ*. If the purpose of VMD is noise reduction, using the quadratic penalty only and keeping the value of *λ* at 0 by setting *τ* to 0. *ε* is typically around 10^−7^. Based on the above analysis, the specific steps for determination of *K* are listed as below:

**Step 1.** Initialization parameters: *K_temp_* = 2, *α* = 2000, *τ* = 0, *ε* = 10^−7^, *num* = 0.

**Step 2.** Perform VMD on raw signal and get IMFs.

**Step 3.** If ADCF occurs, go to step 4, otherwise *num* = 0 and go to step 5.

**Step 4.** *num* = *num* + 1. If *num* is 2, that means ADCF occurs twice continuously, go to step 6, otherwise go to step 5.

**Step 5.** *K_temp_* = *K_temp_* + 1, go to step 2.

**Step 6.** Number of optimized modes *K* = *K_temp_* − 2.

#### 3.1.2. Selection of IMFs

The goal of the K-adaptive VMD is to get an optimal *K* as big as possible, but a big value of *K* will increase the difficulty of component selecting. So component selecting is very important for the proposed K-adaptive VMD. We introduced a component selecting index *En* based on center frequency band (CFB), frequency spectrum energy of original signal after autocorrelation operating and prior knowledge in this subsection. The amplitude of the frequency spectrum of autocorrelation operation is abbreviate as *amp_corr_*. The specific calculation steps are listed as below:

**Step 1.** Calculate autocorrelation operation $R_{x}\left(\tau \right)$ of original signal x(t) by Equation (12) and perform fast Fourier transform to work out the autocorrelation operation frequency spectrum, *i* = 1.


(12)
$$R_{x}\left(\tau \right)=\underset{T\to \infty }{\lim}\int_{0}^{T}x\left(t\right)x\left(t+\tau \right)dt$$


**Step 2.** Compute CFB*_i_* of IMF*_i_*. CFB*_i_* is centered on the center frequency *ω_i_* of IMF*_i_*, and the energy of IMF*_i_* frequency spectrum in CFB*_i_* is *b* times of the total energy of this component.

CFB*_i_* = [*ω_i_* − *f_i_*, *ω_i_* + *f_i_*]
(13)

where *f_i_* is half of CFB*_i_*. The energy ratio *b* is between 0 and 1. The larger the value of *b*, the wider the CFB. In order to get a more concentrated frequency band, this paper takes the energy ratio *b* as 0.3.

**Step 3.** Calculate the energy *En*(*i*) of the *amp_corr_* for raw signal in CFB*_i_*.

**Step 4.** Repeat steps 2 and 3 to get CFB*_i_* (*i* = 1, …, *K*) and *En* = [*En*(1), *En*(2), …,*En*(*K*)] of all IMFs.

**Step 5**. Determine whether prior knowledge is needed to assist in selecting IMF. If so, update *En* according to the knowledge and then go to step 6, otherwise go to step 6.

**Step 6.** Select all components corresponding to *En* index greater than 0.03 × ∑*En*, and reconstruct the signal.

Two aspects of prior knowledge are employed for IMF selecting supplementary when the method K-adaptive VMD is applied to reduce the noise of rolling bearing signals:Prior knowledge one: delete the low-frequency components. Bearing vibration signals frequency bands can be divided into low-frequency, mid-frequency, and high-frequency in frequency domain. Low-frequency signals are usually components related to rotating speed [26]. These low-frequency components will affect the noise reduction effect of K-adaptive VMD. Therefore, the low-frequency components should not be contained in when reconstructing the signal by IMFs, so we can ignore these IMFs by setting *En* value of the components in low frequency to 0. We set *En* value of the IMF1 to IMF[*K*/3] components to 0 (IMF is sorted according to its center frequency from low to high, and [•] means rounded to 0).Prior knowledge two: delete the components containing interference frequencies. In the actual engineering environment, bearing vibration signal is often interfered by natural frequency and rotation frequency of the equipment. In order to remove the influence of these interference frequencies, it is necessary to set *En* corresponding to the CFBs with these interference frequencies to 0.

### 3.2. RBF-FuzzyEn

Fuzzy entropy can represent the complexity of data. Normal bearing vibration signal often present relatively random distribution in amplitude value. The uncertainty of the change for normal bearing vibration amplitude is remarkable, which leads to a big value of fuzzy entropy. When the bearing fault occurs, the distribution of vibration signal amplitude in time domain become more certain, that is, the complexity of signal reduces and the fuzzy entropy is smaller. Therefore, fuzzy entropy of vibration signal in time domain can effectively characterize the state of bearing.

The key to the calculation of fuzzy entropy is fuzzy function. The limitations that need to be met by the fuzzy function have been introduced in Section 2.2. Kernel of radial basis function, which is commonly used in classification algorithms, satisfies the limitations of fuzzy function. The formula of RBF kernel is presented in Equation (14).
(14)$$K\left(x_{i},x_{j}\right)\;=\;\exp\left(-∥x_{i}-x_{j}{∥}^{2}/\left(2\sigma^{2}\right)\right)$$
where $∥x_{i}-x_{j}∥$ is a measure of the distance between ***x****_i_* and ***x****_j_*. Similarly, $d_{ij}^{m}$ in fuzzy entropy is the distance between the coarse-grained sequences *X_i_* and *X_j_*. Hence the $∥x_{i}-x_{j}∥$ in Equation (14) can be replaced by $d_{ij}^{m}$ to form a new fuzzy function based on RBF kernel. Equation of the new fuzzy function is as followings:(15)$$f\left(d_{ij}^{m}\right)=\exp\left(-{\left(d_{ij}^{m}\right)}^{2}/\left(2r^{2}\right)\right)$$

It is apparent that Equation (15) only has one parameter *r* which determines the waveform of the fuzzy function. Figure 2 shows the curve of fuzzy function $f\left(d_{ij}^{m}\right)$ when *r* is 0.01, 0.10 and 0.50, respectively. According to Figure 2, the curve becomes flatter with the increase of *r*. That is, the smaller *r* value is, and the similarity $D_{ij}^{m}$ is more approximate to 0. So as to affect the overall mean of $\varphi^{m}$, and ultimately affects the value of fuzzy entropy.

For small value of *r*, most of $f\left(d_{ij}^{m}\right)$ is about 0, resulting in these $d_{ij}^{m}$ have almost no effect on $\varphi^{m}$ which means a lot of information about the signal distribution will be lost. For big value of *r*, most of $f\left(d_{ij}^{m}\right)$ is close to 1 which will also lead to the same problem. Thus, it is very important to determine *r* adaptively according to the distribution of signal. We proposed that most of $f\left(d_{ij}^{m}\right)$ should not close to 0 based on the abovementioned analysis. To achieve this goal, we need to determine a critical threshold of $d_{ij}^{m}$ that satisfies the following limitations for the reason that $f\left(d_{ij}^{m}\right)$ is a monotone decreasing function: most of $d_{ij}^{m}$ are small than this critical threshold; the fuzzy function valued of this critical threshold is close to 0. Firstly, $\mu \left(d_{ij}^{m}\right)+3\sigma \left(d_{ij}^{m}\right)$ is taken as the critical threshold according to these two limitations. Furthermore, $f\left(\mu \left(d_{ij}^{m}\right)+3\sigma \left(d_{ij}^{m}\right)\right)\;$ is equal to 10^−6^ (it is considered that the value is close to 0 when the value is less than 10^−6^). According to Equation (15) and $f\left(\mu \left(d_{ij}^{m}\right)+3\sigma \left(d_{ij}^{m}\right)\right)\;=\;10^{-6}$, the formula of *r* can be deduced as follows:(16)$$r=\sqrt{\frac{\mu \left(d_{ij}^{m}\right)+3\sigma \left(d_{ij}^{m}\right)}{12\ln10}}$$
where $\mu \left(d_{ij}^{m}\right)$ and $\sigma \left(d_{ij}^{m}\right)$ are the mean and the standard deviation of $d_{ij}^{m}$, respectively. For the calculation of multiple groups of data, *r* is calculated based on the data with the largest $\sigma \left(d_{ij}^{m}\right)$. This is because that the greater the standard deviation of $d_{ij}^{m}$, the more dispersed its distribution which means there are more large $d_{ij}^{m}$. Differ from the existing methods for calculating fuzzy entropy, all data sets adopt a same *r* during the computational process of RBF-FuzzyEn. The specific calculation process of RBF-FuzzyEn is listed hereafter:

**Step 1.** Coarse the original data.

**Step 2.** Calculate the maximum absolute distance $d_{ij}^{m}$ and $d_{ij}^{m+1}$ between *X*(*i*) and *X*(*j*) respectively.

**Step 3.** Compute *r* according to Equation (16).

**Step 4.** Determine the similarity $f(d_{ij}^{m})$ and $f(d_{ij}^{m+1})$ based on Equation (15).

**Step 5.** Calculate fuzzy entropy according to Equations (10) and (11).

## 4. Simulation Analysis and Verification

### 4.1. Analysis and Verification of K-adaptive VMD

#### 4.1.1. Analysis Based on Nonlinear AM-FM Simulation Signal

In this section, a nonlinear AM-FM simulation signals with Gaussian white noise are employed to illustrate the flow and feasibility of the algorithm. The simulation signal *f_sig_*_1_(*t*) is:(17)$$f_{sig1}\left(t\right)=0.6\sin\left(15\pi t+\frac{\pi }{5}\right)+\cos\left(60\pi t+\sin10\pi t\right)+\left(1+0.3\cos\left(10\pi t\right)\right)\sin\left(200\pi t\right)$$
where *t* = 1 s. It can be perceived from Equation (17) that $f_{sig1}\left(t\right)$ consist of three frequency components, which are 7.5 Hz, 30 Hz frequency modulated by 5 Hz and 100 Hz amplitude modulated by 5 Hz, respectively. The sampling frequency is 1k Hz. The SNR of simulation signals is often set within the range of −5~8dB [11,14,18]. We set the Gaussian white noise *η*(*t*) with a SNR of −6 dB to simulate higher noise interference. Figure 3 shows the time domain waveform and frequency spectrum of original signal and noisy signal.

In Figure 3, we can see that noise has strong interference to the original signal in both time domain and frequency domain. The time-domain waveform of original signal has been submerged by noise, and there contains more noises in high frequency range of the frequency spectrum.

In order to verify the K-adaptive VMD method proposed in Section 3.1, *K* is taken as 15, 35 and 150 for VMD decomposition of the AM-FM simulation signal respectively. These three different *K* represent the maximum *K* among the most works of VMD-based bearing fault diagnosis, the optimal decomposition levels obtained by K-adaptive VMD and a very large *K*, respectively. Select the components consistent with the main frequency of original signal when *K* is 13, 35 and 150 respectively, as shown in Figure 4. It can be concluded that with the increase of *K*, the noise in component decreases and all components become smoother by comparing the three subgraphs of Figure 4d–f. This is a manifestation that the band energy of IMF component becomes more concentrated with the increase of *K* value.

As can be seen from Figure 4 that the amplitude modulation component (1+0.3cos(10πt))sin(200πt) of 100 Hz will gradually transform from IMF4 to IMF6 with the increase of *K*; while component of the 7.5 Hz has always been the lowest frequency component; the sideband of 30 ± 5 Hz component will appear in different IMF with the increase of *K*. For example, when *K* = 150, 35 Hz is in the IMF2 and IMF3.

When *K* is equal to 36 and 37, the central frequencies reduce at the high-frequency components as shown in Figure 5b,c, that is, ADCF occurs twice continuously. Therefore, *K* is determined as 35 for AM-FM simulation signal based on the proposed K-adaptive VMD. The phenomenon of ADCF becomes more serious with the increase of *K*, as indicated in Figure 5d. We can observe from the figure that when *K* is big, VMD cannot work out the IMFs with a gentle increase in the center frequency. The center frequency of many components decrease in decomposition results, and the center frequency showed an overall downward trend after about the 100th IMF. These phenomena show that VMD can no longer guarantee a steady increase of IMFs center frequency with the continuous increase of *K* value.

As illustrated in Figure 3f, frequency spectrum of noisy signal has a lot of noise interference. Calculating the *amp_corr_* of original signal can reduce the white noise to a certain extent. Figure 6a shows the frequency spectrum of autocorrelation operation of original signal, several clear peaks correspond to the main frequency of original signal appeared. The mode corresponding to the frequency peak can be accurately selected by calculating the energy value of *amp_corr_* in CFBs. Therefore, the original characteristics of signal are enhanced and the noise is reduced. The energy value *En* is shown in Figure 6b. The red dotted line in the figure is 0.03 × $\sum^{\;}En$. According to this threshold, IMF1, 2, and 6 are selected to reconstruct the signal.

Figure 7 shows the frequency spectrum of reconstructed signal. It can be noticed that noise in the signal reconstructed by K-adaptive VMD is effectively suppressed by comparing Figure 3f and Figure 7a. The noise distributed between 30 Hz and 100 Hz and in the range greater than 100 Hz in the signal has been suppressed to a great extent after noise reduction. For further verification, the simulation signal is also decomposed and reconstructed by frequency band entropy-VMD (FBE-VMD) [10] and tentative VMD (TVMD) [12]. FBE-VMD, TVMD and K-adaptive VMD are comprised of two parts: determination of *K* and component selecting. In the first part for determination of *K*, FBE-VMD takes the kurtosis of IMF’s envelope spectrum as the objective function to optimize *K*, and limits *K* in the range of 2–15. TVMD determines the optimal *K* by comparing the change characteristics of the maximum kurtosis of IMFs when the number of modes is *K* and *K* + 1 respectively. In the second part for component selecting, FBE-VMD operates FBE to judge the fault information in each component. TVMD determines the number of major components according to *K* at first. After that, take the component with the largest kurtosis as optimal component, reckon the distance between optimal component and the other components through dynamic time warping (DTW). The components with smaller distance are selected as the optimal components, and reconstruct the signal by averaging the optimal components. All of the experiments in this paper based on FBE-VMD and TVMD are set as *α* = 2000, *τ* = 0 and *ε* = 10^−7^.

The number of decomposition layers computed by FBE-VMD is 15, and the selected components are IMF11 and IMF13; the optimal *K* estimated by TVMD is 3, and the selected component is IMF2. Figure 7b,c are the frequency spectrum of reconstructed signal decomposed and selected based on FBE-VMD and TVMD, respectively. The conclusion can be drawn that in the signal reconstructed by these two methods, many key frequencies of original signal are not effectively enhanced, but are lost instead. As demonstrated in Figure 7b, none of clear peaks appear at frequencies of 7.5 Hz in sinusoidal signal, 30 ± 5 Hz in FM signal and 100 Hz in AM signal. Moreover, the noise amplitude at high frequencies is relatively high. In Figure 7c, except for the weak amplitude at 100 Hz, the amplitude at other main frequencies is not clear, and noise also dominate the high frequency band. By comparing Figure 7a–c, it can be summarized that noise reduction effect of the proposed method is more pronounced than that of FBE-VMD and TVMD, which not only enhances the spectral amplitude at three groups of main frequencies, but also reduces most of noise.

K-adaptive VMD has great noise reduction effect when the length of simulation signal is 1000. In order to further analyze the noise reduction effect of this method when the sample length changes, five more simulated signal in Equation (17) with 500, 800, 1200, 1500 and 2000 sample length respectively are analyzed. All these data are added with -6dB of noise. The noise reduction results obtained by K-adaptive VMD are shown in Table 1.

We can draw the conclusion that the value of *K* is slightly different when data length changed, but both are greater than 30. Additionally, all of the results have accurately screened out the components corresponding to the three main frequencies. According to the SNR of reconstructed signal, it can be observed that the noise reduction effect is relatively stable with the change of data length, and maintains an excellent level. In the abovementioned results, the SNR of reconstructed signal is the lowest for the group of data with length of 2000, which is 3.77, but it is still significantly improved than −6 dB in the original signal.

#### 4.1.2. Analysis Based on Nonlinear AM-FM Simulation Signal with Close Frequencies

In Section 4.1.1, the three frequencies in the AM-FM signal of are separated. This subsection will analyze the simulated signal $f_{sig2}\left(t\right)$ composed of three components with relatively close frequencies, so as to verify the noise reduction and decomposition ability of K-adaptive VMD when the main frequencies of the signals are relatively close. The simulation signal $f_{sig2}\left(t\right)$ is:
(18)$$f_{sig2}\left(t\right)\;=\;\sin\left(15\pi t+\frac{\pi }{5}\right)+\cos\left(40\pi t+\sin10\pi t\right)+\left(1+0.3\cos\left(10\pi t\right)\right)\sin\left(60\pi t\right)+\eta_{2}\left(t\right)$$
where *t* = 2 s. It can be perceived from Equation (18) that $f_{sig2}\left(t\right)$ consist of three frequency components, which are 7.5 Hz, 20 Hz frequency modulated by 1 Hz and 30 Hz amplitude modulated by 1 Hz, respectively. Similarly, −6 dB Gaussian white noise also added in $f_{sig2}\left(t\right)$. Figure 8 shows the waveforms of $f_{sig2}\left(t\right)$.

The noised signal $f_{sig2}\left(t\right)$ is adaptively decomposed into 37 components by K-adaptive VMD. The corresponding change in the center frequency of IMFs and distribution of *En* is shown in Figure 9.

It is clear in Figure 9b that the optimal IMFs obtained by K-adaptive VMD is IMF1, 2 and 3. The time domain waveforms of these three optimal components are shown in Figure 10a–c, in which the blue dotted line is the original signal waveform corresponding to the IMF. It can be summarized that these three components have good frequency restoration to the original signal. Furthermore, the waveform of IMF1 is very close to the sinusoidal waveform of the original signal. Although part of the amplitude of IMF2 is different from its corresponding FM component, the frequency change is consistent with the original signal, which is more important for FM signals. The amplitude variation trend of IMF3 is consistent with the AM signal as depicted in Figure 10c. Figure 10d is the frequency spectrum in double logarithmic coordinate of the reconstructed signal obtained by summing IMF1, 2 and 3. The amplitude of the reconstructed signal is very obvious at 7.5 Hz, 20 Hz and 30 Hz, meaning that noise in the signal is effectively suppressed. Additionally, FBE-VMD and TVMD are also used for noise reduction, and the results are shown in Figure 10e,f, respectively. The optimal number of *K* obtained by FBE-VMD according to the component kurtosis is 15, which is the maximum value of the optimization range defined by this method. In fact, when *K* gradually increases in the range of 2~15, the maximum kurtosis of the component shows an upward trend. If FBE-VMD does not predetermine the range of *K*, the optimal decomposition layers of VMD is likely to be greater than 15. After decomposition, IMF5, 6, 7, 11, 13 and 15 are screened out according to FBE. The spectrum of reconstructed signal is shown in Figure 10e. It can be observed that none of the three frequencies has a clear amplitude, and the reconstructed signal still contain many noise. TVMD decomposes the original signal into three components adaptively, and selects IMF3 as the optimal component. As shown in Figure 10f, TVMD also does not achieve effective noise reduction.

#### 4.1.3. Analysis Based on Rolling Bearing Fault Simulation Signal

In this section, we will verify the noise reduction effect of K-adaptive VMD through a set of simulation signals of rolling bearing with inner race fault. The formula of simulation signal is:(19)$$\{\begin{array}{c} x\left(t\right)=s\left(t\right)+n\left(t\right)=\underset{i}{\sum }A_{i}h\left(t-iT\right)+n\left(t\right)\\ h\left(t\right)=\exp\left(-Ct\right)\cos\left(2\pi f_{n}t\right)\\ A_{i}=1+A_{0}\cos\left(2\pi f_{r}t\right) \end{array}$$
where *s*(*t*) is the periodic impulse component; amplitude *A*_0_ = 0.3; rotation frequency *f_r_* = 30 Hz; attenuation coefficient *C* = 700; resonance frequency *f_n_* = 4k Hz; ball pass frequency on inner race BPFI = 1/*T* = 120 Hz; *n*(*t*) is Gaussian white noise component with SNR = −13 dB; sampling frequency is set to 16 kHz; length of data is 4096.

The time domain and frequency waveform of impulse signal *s*(*t*) and inner race fault simulation signal *x*(*t*) are shown in Figure 11. Comparing Figure 11a,b, it is found that periodic pulse in simulation signal is completely submerged by noise, and fault characteristics cannot be found through time domain signal. Figure 11c shows the envelope spectrum of noised simulation signal *x*(*t*), and no prominent frequency components can be found. Besides, there is no clear peak at BPFI in the envelope spectrum (0–500 Hz) of Figure 11d.

By using the K-adaptive VMD, 35 is determined as the optimal *K* for rolling bearing inner race fault signal decomposition. The curve of central frequencies of all IMFs is displayed in Figure 12a. After acquiring the IMFs, *En* is calculated through the proposed IMF selecting method, the detail value of *En* can be found in Figure 12b. Since the signal in this section is a bearing fault signal, it is necessary to adopt the first item of prior knowledge mentioned in Section 3.1.2. The composition of simulation signal in this section is clear and there is no obvious interference frequency, so a priori input of interference frequency is not needed. In summary, the input of prior knowledge needs to set *En* corresponding to the 1st to 11th components to 0, and updated *En* is shown in Figure 12c. Same as Figure 6b, the red dotted line in Figure 12c is set at 0.03 × ∑*En*, and the blue column in histogram implies that the optimal IMFs with *En* value greater than the threshold. That is, the optimal components are selected as IMF15, 16, and 26 for reconstruction. The CFBs of IMF15 and 16 are 3921.88~3960.94 Hz and 4078.12~4109.37 Hz, respectively. CFBs of these two IMFs are the closest frequency bands to the resonance frequency 4k Hz in all IMFs. As can be seen from Figure 12c, *En* of these two components near the resonance frequency are much higher than other values.

The reconstructed signal will be acquired after summing these three components. The envelope spectrum of reconstructed signal is shown in Figure 13a, in which clear peaks at BPFI and 2 × BPFI can be recognized. These are significant signs for inner race fault of rolling bearing. Through the comparison of signal envelope spectrums before and after noise reduction, it can be encapsulated that K-adaptive VMD has good noise reduction ability. At the same time, FBE-VMD and TVMD are introduced for the noise reduction effect comparison with K-adaptive VMD. IMF6, 8 and 9 are taken as the optimal components from fourteen IMFs by FBE-VMD. An optimal *K* of 5 is estimated by TVMD, and IMF3 and 4 are selected to reconstruct the new signal. Figure 13b,c are the envelope spectra of the reconstructed signals worked out by these two methods, respectively. There is no clear peak at fault characteristic frequency in Figure 13b, and the noise reduction effect is poor. Even though clear peaks at BPFI and 2 × BPFI appear in Figure 13c, there are more interference frequencies in the frequency band of 300–500 Hz. The envelope spectrum component of signal computed by K-adaptive VMD is simpler. Additionally, the amplitude value of BPFI in Figure 13a is much larger than that of Figure 13c.

#### 4.1.4. Verification of Noise Reduction Effect

In this section, different intensity of noise are added to the above two groups of simulation signals, and the K-adaptive VMD, FBE-VMD and TVMD algorithms are brought in to decompose and reconstruct simulation signal to verify noise reduction ability of these three methods under different noise levels.

Add noise to the to the AM-FM signals described in Section 4.1.1 to get noisy signal with 6 and −6dB SNR, respectively. Table 2 shows the noise reduction results of those three VMD methods. All experiments are conducted in the MATLAB (R2020b) installed in high performance computing platform of Beijing Jiaotong University with 48 core CPU. From the SNR of reconstructed signal in Table 2, K-adaptive VMD has the best noise reduction effect on these two signals with different noise levels. As revealed in Table 2, the *K* computed by K-adaptive VMD is 17 and 35 respectively, while the *K* obtained by the other two methods are 12, 15 and 3, 3, respectively. Because we do not set an upper limit of *K*, the optimal *K* reckoned by K-adaptive VMD is bigger than those of the comparison methods. This illustrate that setting an upper limit of *K* may not necessarily for optimizing it. We can draw the conclusion based on CFBs of the selected IMFs that K-adaptive VMD can accurately select the components close to the three main frequencies of simulation signal at 7, 30, and 100 Hz. However, central frequencies of the selected components using FBE-VMD and TVMD are quite different from those three main frequencies. Since FBE-VMD needs to manually filter optimal IMFs according to the change of the FBE, and does not realize adaptive decomposition and reconstruction, its running time cannot be counted, and the method is the most complicated. It can also be easily concluded that, K-adaptive VMD takes more time in two signals than TVMD and has a medium complexity. Although K-adaptive VMD has certain computational burden, the high performance computing platform widely used nowadays can be very good solution to his problem.

A noisy signal of inner race bearing fault is added on the basis of the simulation signal in Section 4.1.3 to verify the performance of three methods when the vibration signal of rolling bearing is at different noise levels. Since the noise content of the simulation signal in Section 4.1.3 is −13 dB and the SNR is relatively low, the SNR of the new signal is set slightly higher, which is −6 dB. The above three methods are utilized to reduce noise of these two sets of rolling bearing inner race fault simulation signals with different noise, and the results are shown in Table 3. Similar to the results of AM-FM signal, the *K* optimized by K-adaptive VMD is much bigger than those of the other two methods. It is clear in Table 3 that the noise reduction performance of the proposed method is much better than that of FBE-VMD. For the signal with −6 dB noise, although the SNR after noise reduction using K-adaptive VMD is slight lower than TVMD, the noise has also been significantly suppressed. For signals with higher noise level, SNR = −13 dB, K-adaptive VMD achieves better noise suppression effect than TVMD.

### 4.2. Analysis and Verification of RBF-FuzzyEn

The purpose of this section is to verify the influence of noise and data length of bearing vibration signal in different fault types on feature discrimination of RBF-FuzzyEn. On the basis of inner race bearing fault signal in Section 4.1.3, an outer race bearing fault simulation signal is added in this section, and the formula is as follows:(20)$$\{\begin{array}{c} x\left(t\right)=\underset{i}{\sum }Ah\left(t-iT\right)+n\left(t\right)\\ h\left(t\right)=\exp\left(-Ct\right)\cos\left(2\pi f_{n}t\right) \end{array}$$
where amplitude *A* = 0.3; rotation frequency *f_r_* = 30 Hz; attenuation coefficient *C* = 200; resonance frequency *f_n_* = 4 kHz; ball pass frequency on outer race BPFO = 1/T = 80 Hz; sampling frequency is set as 16 kHz; the number of analysis points is 4096; and *n*(*t*) is Gaussian white noise.

#### 4.2.1. Analysis of the Distribution for Maximum Absolute Distance

The maximum absolute distance *d_ij_* is the independent variable of fuzzy function which directly determines the value of fuzzy entropy. Thus, understanding the distribution of *d_ij_* is of great consequence for the analysis of RBF-FuzzyEn.

Firstly, we analyze the *d_ij_* characteristics of inner race fault impulse signal. Compute the distance matrix $d_{ij}^{m}$ and $d_{ij}^{m+1}$ of the periodic fault impulse signal *s*(*t*) based on Equation (8). The number of points for $d_{ij}^{m}$ calculated from each set of data is (*N* − *m*) × (*N* − *m* + 1), where *N* is the length of data. The simulation signal employed in this section is 4096 points, so the number of $d_{ij}^{m}$ and $d_{ij}^{m+1}$ are 16764930 and 16756742, respectively. The values of $d_{ij}^{m}$ and $d_{ij}^{m+1}$ are shown in Figure 14. Figure 14a,b show all the *d_ij_*. The abscissa is (*i* = 1, *j* = 2~*N* − *m* + 1), …, (*i* = *N* − *m* + 1, *j* = 1~*N* − *m*) successively, and ordinate represents the corresponding maximum distance value. Figure 14c,d show *d_ij_* when *i* = 1 and *j* = 2~*N* − *m* + 1. Figure 14a,b reveal that *d_ij_* also has the same periodic characteristics as fault impulse in the whole range which we will call it big cycle. In addition, *d_ij_* shows the same cycle as the fault impulse in a small range from inspecting Figure 14c,d, which can be called small cycle. Moreover, whether *d_ij_* is in big cycle or small cycle, the amplitude modulation phenomenon appears, which share the same characteristics with inner race fault signal.

For the purpose of observing the distribution of maximum absolute distance matrix *d_ij_* for bearing signals in two different states, K-adaptive VMD is implemented to reduce the noise of these two groups of signal. Then extract the fuzzy entropy feature from denoised signal. The points of *d_ij_* for the two groups of data are the same. The number of $d_{ij}^{m}$ is 16,764,930, and the number of $d_{ij}^{m+1}$ is 16,756,742. The *d_ij_* is counted at a resolution of 2000 points to figure out the distribution of *d_ij_* for two states, as presented in Figure 15. The abscissa means the value of $d_{ij}^{m}$ or $d_{ij}^{m+1}$ for inner race fault and outer race fault. The ordinate represents the total numbers of *d_ij_* which is abbreviated as *Numd*.

As can be noticed from Figure 15, the distance matrix distribution of the bearing signals in two states are quite different. The distance distribution of outer race fault is more scattered than that of inner race fault, indicating that the self-similarity of outer race fault data is lower than that of inner race fault data. For outer race fault signal, the distribution curve of distance matrix is smoother than inner race fault signal no matter for $d_{ij}^{m}$ or $d_{ij}^{m+1}$. The standard deviation of these two signals’ $d_{ij}^{m}$, 0.18 and 0.25 respectively, indicating that $d_{ij}^{m}$ of the outer race fault signal has more dispersed distribution. For the same state, the distance distribution when *m* = 3 is more dispersed than *m* = 2. It can also be revealed from the statistical figure that the maximum *Numd* of $d_{ij}^{m}$ is close to zero. Thus, the *Numd* of $d_{ij}^{m}$ decline gradually for both inner and outer race fault data. While, the distribution curves for $d_{ij}^{m+1}$ show an upward trend and then a downward trend. Based on the above analysis, the distance *d_ij_* of bearing vibration signals under various fault types shows quite different distribution. Therefore, a constant *r* value, that is, the same similarity function, can be executed to estimate the entropy value to characterize the states of different data. According to *d_ij_* properties of the above two state data, it can be noticed that the distribution of outer race fault data is broader. Therefore, *r* is calculated by $d_{ij}^{m}$ of the outer race fault simulation signal, and *r* = 0.2065 is worked out.

#### 4.2.2. Analysis and Verification under Different Noise Level

We add noise to inner and outer race fault simulation signals, with the SNR ranging from 20dB to -20dB with step -1dB to evaluate the relationship between RBF-FuzzyEn and noise. Besides RBF-FuzzyEn, two different fuzzy entropy calculation methods [23,24], approximate entropy (ApEn) [21], sample entropy (SampEn) [22] and multiscale fuzzy entropy (MFE) [27] are compared at the same time. The fuzzy function in paper [23] is $f_{1}\left(d_{ij}^{m}\right)=\exp\left(-{\left(d_{ij}^{m}\right)}^{n}/r\right)$, and in [24] is $f_{2}\left(d_{ij}^{m}\right)=\exp\left(-{\left(d_{ij}^{m}/r\right)}^{n}\right)$. Compared with the function proposed in this paper, there is one more parameter *n* in these two fuzzy functions. According to papers [23,24], *m* = 2, *n* = 2, and *r* = 0.2 × *σ* (data). FuzzyEn1 and FuzzyEn2 are applied to represent the entropy worked out by methods in paper [23,24], respectively. ApEn and SampEn both have the parameters of *m* and *r*, which are set the same as FuzzyEn1 and FuzzyEn2, that is, *m* = 2, and *r* = 0.2 × *σ* (data). With regard to MFE, we set *m* = 2, *n* = 2, *r* = 0.15 × σ (data) and the scale factor *τ* = 20 according to paper [27]. Figure 16 shows the distribution of entropy for signals with different noise level. The abscissa donates -SNR, i.e., the noise in simulation signal increases along the positive direction of abscissa.

For RBF-FuzzyEn and FuzzyEn1, the entropy differentiation between inner race and outer race fault signals is remarkable despite the increase of noise as described in Figure 16a,b In addition, the two curves of inner and outer race fault data maintain the same trend while SNR is reducing. Moreover, RBF-FuzzyEn achieve the most apparent differentiation when noise content stay at the same level by comparing with FuzzyEn1. The entropy difference between inner and outer race fault signals under different noise levels are calculated. The average value of them for RBF-FuzzyEn and FuzzyEn1 are 0.4826 and 0.3480, respectively, which means that the differentiation of RBF-FuzzyEn is better than FuzzyEn1. Different from Figure 16b, the change of RBF-FuzzyEn is small with the decrease of SNR. It is clear in Figure 16c that FuzzyEn2 of the two fault types signals close to each other gradually with the increase of noise content in the signal, blurring the distinction between two states. And the curves begin to overlap when SNR is less than −6dB, demonstrating that FuzzyEn2 is susceptible to noise. The entropy curves of ApEn and SampEn are very similar, and the entropies of different states are almost mixed together. Only when SNR is within the range of 0~10 dB, the entropy obtained by ApEn and SampEn has small distinction between inner and outer race faults. With regard to MFE, the difference between two states is very small, which can not effectively represent the bearing state. All the above comparisons reach a conclusion that the RBF-FuzzyEn possesses strong noise robustness.

#### 4.2.3. Analysis and Verification with Various Data Length

Fuzzy entropy has the advantage of low dependence on data length *N*. To analyze and verify the relationship between RBF-FuzzyEn and data length, this section generates fault simulation signals of inner and outer race fault of rolling bearings with different data lengths based on Equations (19) and (20). The data length *N* range from 800 to 8000 with a step length of 800. Each state is composed of 10 groups of data, a total of 20 groups of data. Noise is added to each group of data, making the SNR of all data set equal to −13dB. First, K-adaptive VMD is implemented on these 20 groups of data to reduce the noise to some extent. Then, calculating fuzzy entropy of the denoised signal based on RBF-FuzzyEn, Fuzzyen1, Fuzzyen2, ApEn, SampEn and MFE method successively. Figure 17 shows the variation of entropy of five calculation methods when *N* takes different values.

As depicted in Figure 17, the entropy values estimated though RBF-FuzzyEn and FuzzyEn1 show good stability with the change of data length *N*. And the distinction of fuzzy entropy calculated based on data in different states is always significant. However, the entropy of inner and outer race fault data computed by FuzzyEn2, ApEn and SampEn vary greatly. Furthermore, the entropy of two states are very close when *N* is 800 and 1600 for FuzzyEn2. As for ApEn, the entropy can be hardly distinguished under 1600 length of data. Although the MFE curve is very stable with the change of N, the two curves almost overlap in the whole range of N and the feature discrimination is very poor. The mean values of fuzzy entropy difference between inner and outer race faults are 0.4534, 0.3253, 0.0334, 0.0456, 0.0484 and 0.0085, respectively. Obviously, RBF-FuzzyEn has higher feature discrimination than FuzzyEn1, and both of them are much greater than FuzzyEn2, ApEn, SampEn and MFE. To sum up, RBF-FuzzyEn has the advantages of low data length dependence and excellent distinction compared with FuzzyEn1, FuzzyEn2, ApEn, SampEn and MFE, which is more suitable to distinguish the bearing state by experimental vibration signal.

## 5. Application in Bearing Feature Enhancing

### 5.1. Case 1: Bearing Data of CWRU

The experimental data of rolling bearing failures introduced in this section are derived from the bearing dataset of Case Western Reserve University (CWRU) [28]. The sampling frequency of selected data sets is 12k Hz, and all of them contain three accelerometers which are located on motor drive end, motor fan end and bench base. Signal collected by sensor located on the base of bench contains high noise due to the influence of transmission path and complicated working environment. Compared with the signal collected from drive end and fan end, the base sensor signal has lower SNR, making noise reduction more difficult. In this section, two sets of base signals in drive end bearing fault dataset are selected to verify the noise reduction performance of K-adaptive VMD. Details of the data are itemized in Table 4.

Each group of data was analyzed with 6000 points, that is, 0.5 s. Figure 18 shows the *amp_corr_* of data set 169 and 158. The frequency bands indicated by red box in Figure 18a,b contain clear frequency peaks located at 4140 Hz along with sidebands at intervals of 60 Hz which is twice of motor rotational frequency. Thus, these frequencies appearing in the experiments of different bearing states and different working conditions are comprised by motor rotation frequency. Because the peaks at these interference frequency are apparent, the corresponding *En* of CFBs containing these frequency components will also be considerable when calculating the energy of *amp_corr_* in CFB. This will result in failure to select the components containing fault information. Based on this prior knowledge, *En* corresponding to these interference frequencies should be set to 0 when the method proposed in this paper is applied to experimental signal of case 1.

#### 5.1.1. Noise Reduction

Figure 19 shows time domain waveform and envelope spectrum of the data set 169. There constitute no noticeable characteristics of bearing inner race fault in both time domain diagram and envelope spectrum.

Analyze data through K-adaptive VMD, the dataset 169 is decomposed into thirteen IMFs. The time domain signal of all IMFs is presented in Figure 20. Table 5 lists the CFBs of all IMFs. The low-frequency components in these 13 IMFs are IMF1~3. The time domain waveform of IMF3 has very obvious impulse which appeared15 times in the 0.5 s signal, corresponding to the rotational frequency. The amplitude changes of IMF9 and IMF10 in time domain are relatively similar. All of them is about 60 Hz, which is twice the rotational frequency. It is also revealed in Table 5 that the CFBs of these two components both contain the interference frequencies analyzed above. Moreover, IMF11 also contains the interference frequencies. In addition, *En* of IMF1-IMF[*K*/3], i.e., *En* of the 1st to the 4th component should also be set to 0 according the prior knowledge one mentioned in 3.1.2.

Set the *En* of IMF1, 2, 3, 4, 9, 10, 11 (as displayed in the red column diagram in Figure 21a) to 0, and get the updated *En* distribution as presented in Figure 21b. According to the updated *En*, IMF5, 6, 7, 8 and 12 are selected.

By summing all the selected IMFs, the reconstructed signal is obtained, and its envelope spectrum is shown in Figure 22a. It is obvious that K-adaptive VMD can suppress noise and enhance fault characteristics effectively by comparing Figure 19b and Figure 22a. Figure 22b shows the envelope spectrum of reconstructed signal worked out by FBE-VMD. The number of modes *K* is 14, and IMF5 and 7 are selected as the optimal components. Figure 22c is the envelope spectrum of reconstructed signal computed by TVMD, with mode number *K* = 3, and IMF2 is chosen as optimal component. Comparing Figure 22a–c, it can be summarized that the noise reduction effect of K-adaptive VMD is better than the other two methods. Although the envelope spectrum of the reconstructed signal acquired by FBE-VMD contains BPFI and corresponding sidebands, peaks located at 2 × BPFI and its sidebands are not existent. In the envelope spectrum of signal reconstructed by TVMD, only BPFI appears with small amplitude, and there is no 2 × BPFI peak and sideband either. While, the envelope spectrum of signal reconstructed by K-adaptive VMD contains remarkable BPFI, 2 × BPFI and corresponding sidebands.

Vibration signal from bench base sensor contained in the 158th data set is taken as example to test and verify the effect of K-adaptive VMD for bearing outer race fault. Figure 23 shows time domain waveform and envelope spectrum of 0.5 s data selected from this data set.

Figure 23 indicated that fault impact are not clear in time domain waveform due to noise. In the envelope spectrum, there exist small peaks at the fault characteristic frequency and its harmonic. Through K-adaptive VMD, the optimal number of decomposition components is determined to be 16 firstly, corresponding time domain waveforms and CFBs are itemized in Figure 24 and Table 6, respectively. The first five components are considered as low-frequency components according to prior knowledge. They are not used to reconstruct the signal because they generally consist of rotational frequency. The time domain waveforms of IMF3 and IMF4 in Figure 24 have relatively similar periodic shocks of about 30 Hz, that is, the rotational frequency. Among the high-frequency components, IMF12 and IMF13 have the same periodicity of 60 Hz, which is consistent with the prior knowledge two put forward in Section 3.1.2 and the interference frequencies in Figure 18b. These analyses both verify the validity of the two prior knowledge in Section 3.1.2.

Based on the above mentioned analysis, the *En* of IMF1~IMF5 and IMF10~14 should also be set to 0 based on prior knowledge. Figure 25 illustrate the *En* value before and after updating. Finally, IMF6, 7, 8, 9 and 15 are selected for reconstruction.

The envelope spectrum of reconstructed signal acquired by summing the above selected components is shown in Figure 26a. Clear peaks at BPFO and its 2×~4× harmonics can be easily found in Figure 26a, and interference frequency components have been suppressed comparing with Figure 23b. In FBE-VMD, IMF8, 11, and 12 out of 12 components are chosen for signal reconstruction, its envelope spectrum is present in Figure 26b. Meanwhile, IMF2 is taken as the optimal component in TVMD method, while *K* = 3. Figure 26c shows the envelope spectrum of IMF2 evaluated by TVMD. By comparing Figure 26a–c, the conclusion can be drawn that K-adaptive VMD is superior to FBE-VMD and TVMD in terms of noise reduction and characteristic enhancement performance.

#### 5.1.2. Feature Extraction

In this section, two sets of reconstruction signals estimated by K-adaptive VMD in Section 5.1.1 and a set of normal bearing vibration signal from CWRU dataset are used to test and verify the proposed fuzzy entropy method in feature extraction. Due to bench base data under normal conditions are not applied in the dataset, we select vibration signal from fan end sensor contained in the 97th dataset as an alternative, corresponding rotational speed is 1796 rpm. The length of these three groups of data also be set to 6000 points. We employed N, IF and OF as abbreviations for normal, inner race fault and outer race bearing fault, respectively.

The standard deviations of $d_{ij}^{m}$ for these three data are 0.0159, 0.0112 and 0.0070, respectively. For the reason that the standard deviation of N is the largest among them, $d_{ij}^{m}$ of normal bearing is substituted into Equation (16), and the parameter *r* of similarity function is calculated to be 0.0131. According to Equations (10), (11) and (15), fuzzy entropy of N, IF and OF data can be computed as 1.0523, 0.5963 and 0.4784, respectively. It can be observed that the entropy of normal bearing signal is much bigger than that of fault bearing signal, which conform to the characteristics of entropy and the vibration mechanism of bearing. The spalling of bearing will cause periodic impact, which will enhance the certainty of signal’s distribution in time domain. Therefore, the entropy value of signal decreases. Furthermore, distinction on entropy among three groups of data is also very obvious. For the purpose of verifying effectiveness of RBF-FuzzyEn, this section compared the fuzzy entropy of FuzzyEn1 and FuzzyEn2. The fuzzy entropy computed based on these three methods are recorded in Table 7.

According to the characteristics of five methods, the *r* of each state of data in FuzzyEn1, FuzzyEn2, ApEn, SampEn and MFE is different, and they are all related to the distribution of data. We can observe from Table 7 that, for the same bearing data, FuzzyEn1, FuzzyEn2, ApEn and SampEn share the same *r* value, because all of those methods estimate *r* with the formula of 0.2 × *σ* (data). Nevertheless, RBF-FuzzyEn utilizes the same *r* for different groups of data. It can also be concluded from Table 7 that the entropy values of three states are all around 0.1. Meanwhile, the entropy values between N and IF judged by FuzzyEn1 are very close, with a difference of only 0.0011, resulting in poor distinguishing between different bearing states. Considering FuzzyEn2, the three entropy values are all above 1. Contrary to common sense, fuzzy entropy of faults signals are greater than that of normal signal, which also cause a failure distinguish on bearing states. The method of determining *r* according to the standard deviation in FuzzyEn1 and FuzzyEn2 leads to unique fuzzy function in each group of data. In RBF-FuzzyEn, *r* is judged according to normal bearing data. So, three sets of data share the same *r*, that is, the same fuzzy function. To sum up, RBF-FuzzyEn is superior to the other two methods in terms of bearing states distinguishing ability based on entropy features and good consistency between entropy values and bearing fault state. The results of ApEn and SampEn are similar to FuzzyEn2, and the entropy of bearing fault data is greater than that of normal data. Although the normal bearing entropy obtained by the MFE method is relatively large, the entropy of IF and OF are very close with a difference of only 0.0331, which is much less than 0.1179 of RBF-FuzzyEn.

### 5.2. Case 2: Bearing Data of IMS

The experimental data of rolling bearing in this section comes from the rolling bearing life cycle data set of American NSFI/UCRC Center for Intelligent Maintenance Systems (IMS) [29]. Four bearings are installed on a rotating shaft of the test bench. The rotating speed is constant at 2000 rpm. Each of data set contains one second vibration signal collected at 20k Hz sampling frequency every 10 min. Three times of experiments were carried out. At the end of the second experiment, it is verified that the defect occurred on bearing one outer race, and the corresponding fault characteristic frequency is 236.4 Hz. We select the data at 5410 min in the second experiment for research.

First, we investigate the autocorrelation operation spectrum of signal at 5410 min. It is clear in Figure 27a that peak at 986 Hz, which is neither related to rotation frequency nor BPFO, dominates in the whole spectrum. In order to figure out this phenomenon, signal at 100 min, when bearing is under normal state, is checked by autocorrelation operation and spectral analysis. Once again, as displayed in Figure 27b, peak at 986 Hz has the highest amplitude, which bear no relation to bearing states. Therefore, peak at 986 Hz should be regarded as interference, *En* corresponding to the CFB containing this frequency should be modified to 0 when performing K-adaptive VMD.

We take the first 0.5 s data at 5410 min to analyze. Figure 28 shows time-domain waveform and envelope spectrum of this data. As depicted in Figure 28 that peaks at BPFO and its 2 × −4 × harmonics exist in envelope spectrum, but there are some peaks of noise interference around 2 × BPFO and 3 × BPFO, making the amplitude does not show a decreasing state from 2× to 4 × harmonic.

In order to suppress noise interference and enhance fault characteristic, K-adaptive VMD is put into effect. After determining the optimal *K*, the signal is decomposed into 28 IMFs, their CFBs are itemized in Table 8. According to the prior knowledge, IMF3 contains the 986 Hz interference frequency. In addition, the first [*K*/3] IMFs, that is, the 1st–9th component, are also be considered as interference.

As illustrated in Figure 29a, the original *En* of IMF3 is huge. We update the indicators by modifying the *En* of IMF1-9 to 0. Finally, IMF10, 11, 12, 13, and 14, as the blue column described in Figure 29b, are selected for reconstruction.

Figure 30a shows the envelope spectrum of reconstructed signal. Comparing with Figure 23b, it can be demonstrated that peaks of noise interference around 2 × BPFO and 3 × BPFO is significantly suppressed, and amplitude from 2× to 4 × harmonics present a downtrend. Figure 30b shows the envelope spectrum of signal reconstructed by FBE-VMD, in which IMF13 is selected out of 14 components. Although peaks at BPFO and its harmonics appear in spectrum, they experience similar interference in the envelope spectrum of the original signal. In TVMD, IMF2 is chosen from three IMFs and its envelope spectrum is illustrated in Figure 30c. Peaks of fault characteristic can be only found at BPFO and 2 × BPFO, and the amplitude of peaks are not very clear. The conclusion can be drawn by comparing Figure 30a–c that the K-adaptive VMD possesses satisfactory noise suppression and fault feature enhancement performance in processing bearing vibration signal.

## 6. Conclusions

In this paper, VMD and fuzzy entropy are improved from aspects of noise reduction and feature extraction respectively, forming a feature enhancement method that combines the K-adaptive VMD and RBF-FuzzyEn for the fault diagnosis of bearing. The main contributions of this paper can be summarized as below:

An algorithm of K-adaptive VMD was developed, which can obtain *K* adaptively based on ADCF, and select the optimal IMFs by a coefficient *En*. The analysis of reconstructed signal of five types of signals (nonlinear AM-FM simulation signal, nonlinear AM-FM simulation signal with close frequencies, inner race fault simulation signal, CWRU rolling bearing inner and outer race fault signal and IMS bearing outer race fault signal) demonstrated that compared with FBE-VMD and TVMD, the K-adaptive VMD has good noise reduction ability. The envelope spectrum of simulation and experimental bearing fault signals proved that K-adaptive VMD can realize the effective extraction of fault impulses.RBF-FuzzyEn was proposed, which introduced an innovative fuzzy function and carried out a specific way for determination of parameter *r* in fuzzy function. To verify the entropy feature extraction method proposed in this paper, RBF-FuzzyEn, FuzzyEn1, FuzzyEn2, ApEn, SampEn and MFE are used to extract the entropy of bearing fault simulation signals under different SNR and data length, as well as bearing fault experiment signals. The conclusion can be obtained from the results that RBF-FuzzyEn outperformed FuzzyEn1, FuzzyEn2, ApEn, SampEn and MFE in distinction between different sates of bearing. Meanwhile, the proposed RBF-FuzzyEn shows a marked noise robustness than other entropy methods and it is independent on data length *N*.

## Figures and Tables

**Figure 1 entropy-24-00197-f001:**
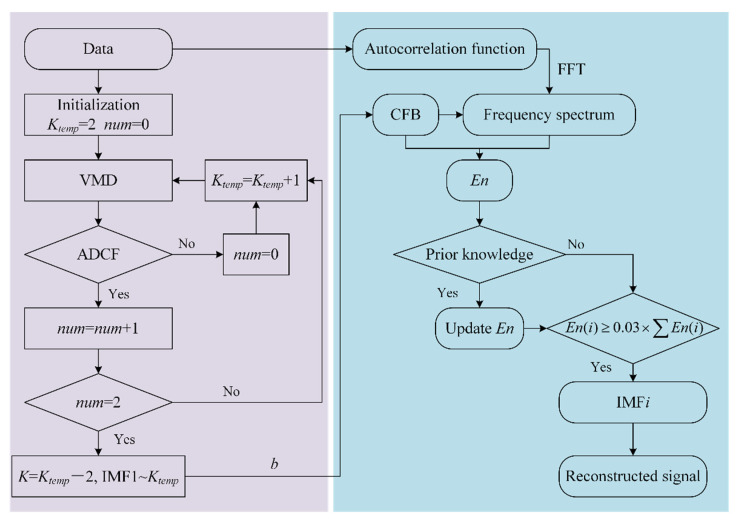
Flowchart of K-adaptive VMD.

**Figure 2 entropy-24-00197-f002:**
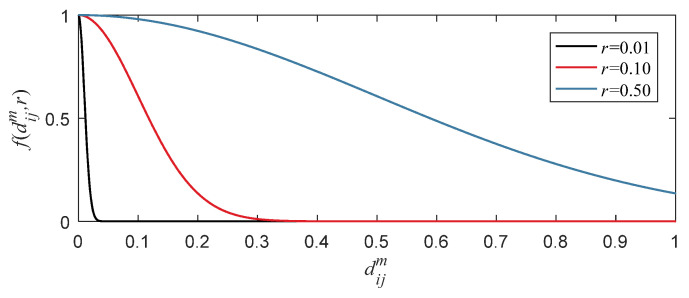
Fuzzy function curve (*r* is equal to 0.01, 0.10 and 0.50, respectively).

**Figure 3 entropy-24-00197-f003:**
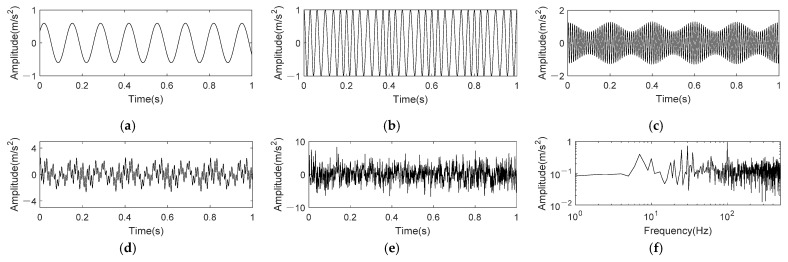
Time domain and frequency spectrum of nonlinear AM-FM signal. (**a**) $0.6\sin\left(15\pi t+\frac{5}{\pi }\right)$; (**b**) cos(60πt+sin10πt); (**c**) (1+0.3cos(10πt))sin(200πt); (**d**) Original signal; (**e**) Noisy signal; (**f**) Frequency spectrum of noisy signal.

**Figure 4 entropy-24-00197-f004:**
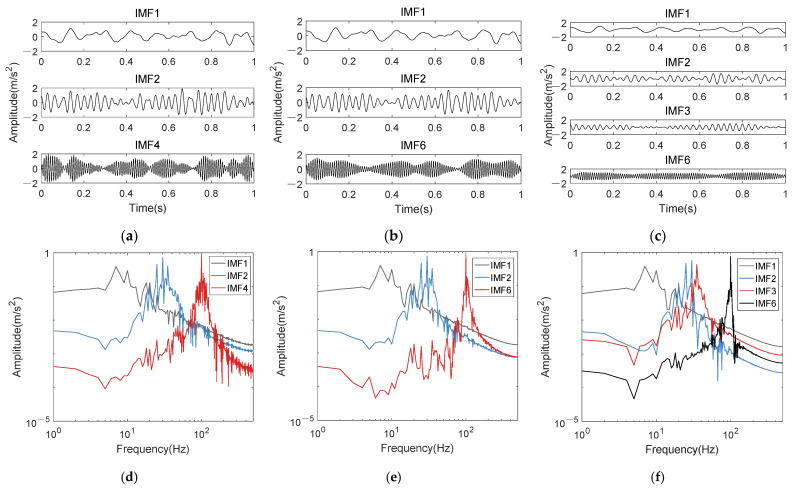
Components consistent with the main frequency of the original signal. (**a**) IMF1, 2 and 4 when *K* is taken as 15; (**b**) IMF1, 2 and 6 when *K* is taken as 35; (**c**) IMF1, 2, 3 and 6 when *K* is taken as 150; (**d**–**f**), frequency spectra of (**a**–**c**).

**Figure 5 entropy-24-00197-f005:**
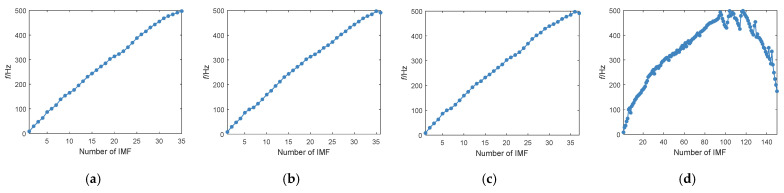
Change of center frequency for IMFs with different *K*. (**a**) *K* = 35; (**b**) *K* = 36; (**c**) *K* = 37; (**d**) *K* = 150.

**Figure 6 entropy-24-00197-f006:**
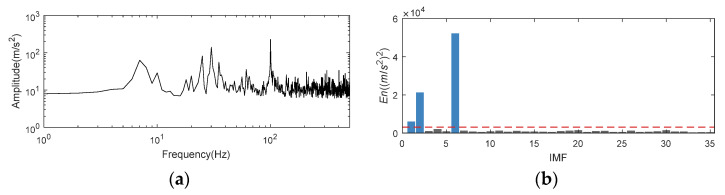
Frequency spectrum of autocorrelation operation and *En* for nonlinear AM-FM simulation signal. (**a**) Frequency spectrum of autocorrelation operation in logarithmic coordinates; (**b**) *En* obtained by K-adaptive VMD.

**Figure 7 entropy-24-00197-f007:**
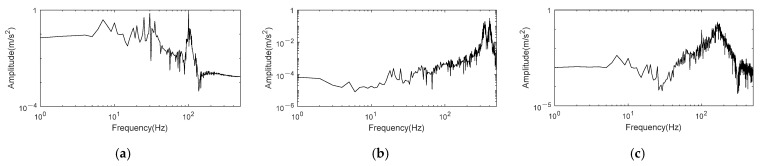
Frequency spectrum of reconstruction signal for AM-FM signal obtained by three methods. (**a**) Reconstruction signal obtained by K-adaptive VMD; (**b**) Reconstruction signal obtained by FBE-VMD; (**c**) Reconstruction signal obtained by TVMD.

**Figure 8 entropy-24-00197-f008:**
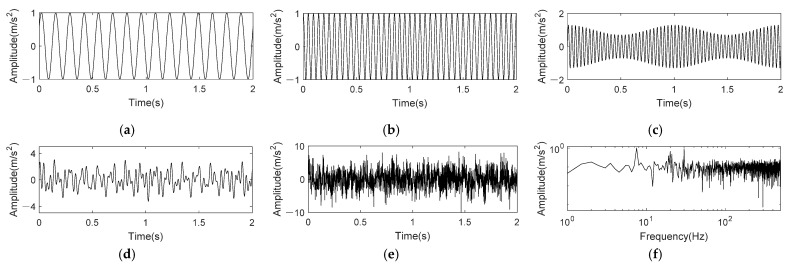
Time domain and frequency spectrum of nonlinear AM-FM with close frequencies. (**a**) $\sin\left(15\pi t+\frac{\pi }{5}\right)$; (**b**) cos(40πt+sin10πt); (**c**) (1+0.3cos(10πt))sin(60πt); (**d**) $f_{sig2}\left(t\right)$; (**e**) Noisy signal; (**f**) Frequency spectrum of noisy signal.

**Figure 9 entropy-24-00197-f009:**
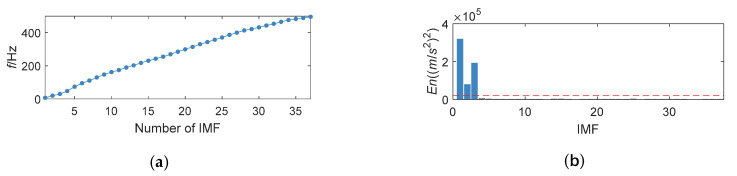
Change of center frequency for IMFs and *En*. (**a**) Change of center frequency for IMFs; (**b**) *En* obtained by K-adaptive VMD.

**Figure 10 entropy-24-00197-f010:**
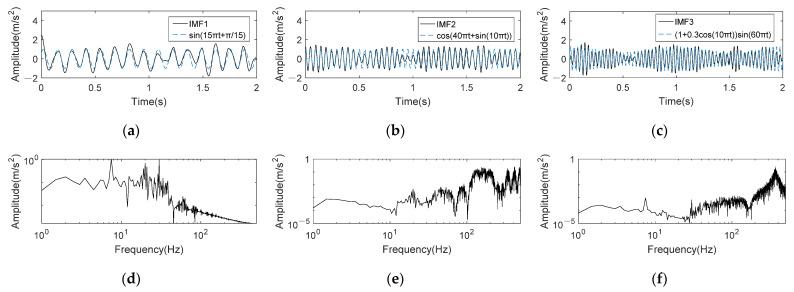
Time domain and frequency spectrum of reconstruction signal for $f_{sig2}\left(t\right)$ obtained by three methods. (**a**–**c**) Time domain waveforms of IMF1~3 obtained by K-adaptive VMD; (**d**) Reconstruction signal obtained by K-adaptive VMD; (**e**) Reconstruction signal obtained by FBE-VMD; (**f**) Reconstruction signal obtained by TVMD.

**Figure 11 entropy-24-00197-f011:**
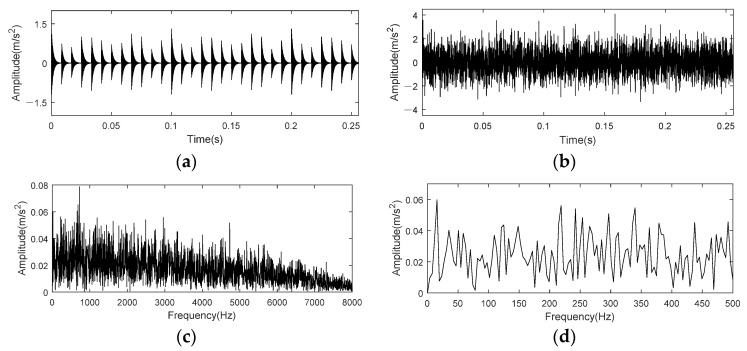
Time domain and frequency spectrum of simulation signal for rolling bearing inner race fault. (**a**) Periodic impulse component; (**b**) Signal with noise; (**c**) Envelope spectrum of noisy signal; (**d**) Envelope spectrum of noisy signal (0–500 Hz).

**Figure 12 entropy-24-00197-f012:**
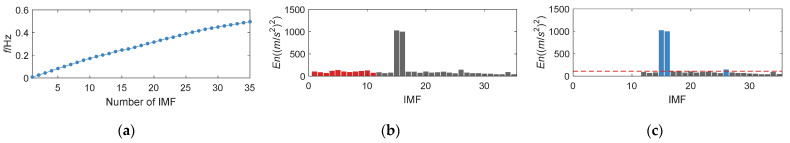
*En* of simulation signal for inner race bearing fault. (**a**) Change of central frequencies; (**b**) Original *En*; (**c**) Updated *En*.

**Figure 13 entropy-24-00197-f013:**
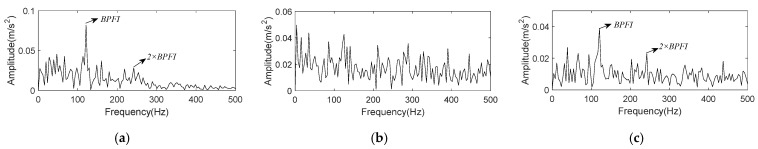
Envelope of reconstruction signal. (**a**) Reconstruction signal obtained by K-adaptive VMD; (**b**) Reconstruction signal obtained by FBE-VMD; (**c**) Reconstruction signal obtained by TVMD.

**Figure 14 entropy-24-00197-f014:**
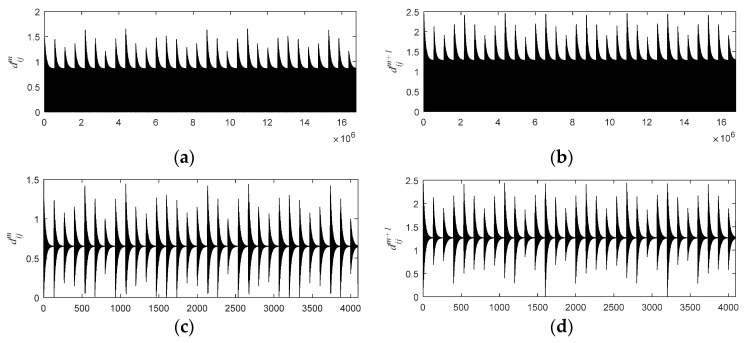
Maximum distance *d_ij_* of inner race fault impulse signal. (**a**) $d_{ij}^{m}\;$ in big cycle; (**b**) $d_{ij}^{m+1}$ in big cycle; (**c**) $d_{ij}^{m}$ in small cycle (*i* = 1, *j* = 2~4095); (**d**) $d_{ij}^{m+1}$ in small cycle (*i* = 1, *j* = 2~4094).

**Figure 15 entropy-24-00197-f015:**
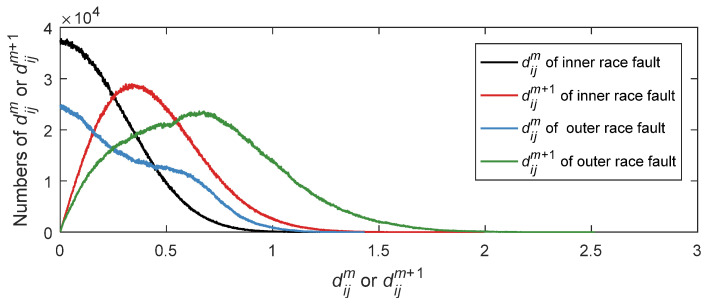
Maximum absolute distance distribution of bearing fault simulation data.

**Figure 16 entropy-24-00197-f016:**
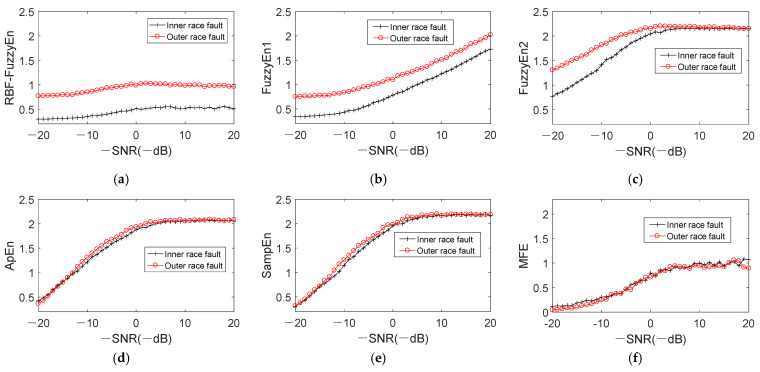
Variation of entropy with different noise. (**a**) RBF-FuzzyEn; (**b**) FuzzyEn1; (**c**) FuzzyEn2; (**d**) ApEn; (**e**) SampEn; (**f**) MFE.

**Figure 17 entropy-24-00197-f017:**
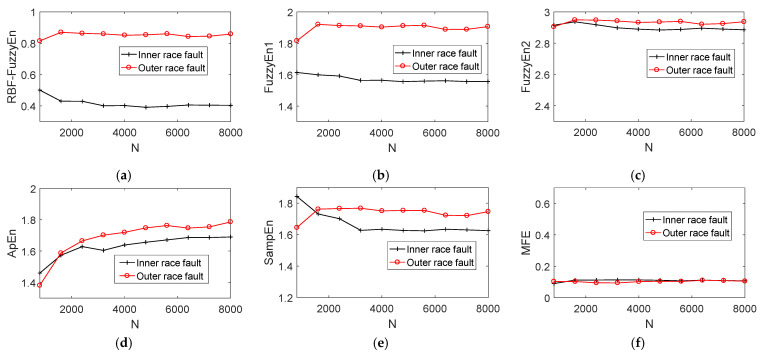
Variation of fuzzy entropy with data length N. (**a**) RBF-FuzzyEn; (**b**) FuzzyEn1; (**c**) FuzzyEn2; (**d**) ApEn; (**e**) SampEn; (**f**) MFE.

**Figure 18 entropy-24-00197-f018:**
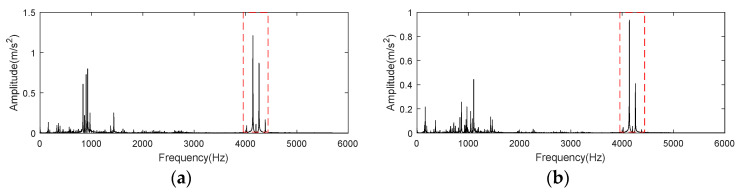
Frequency spectrum of autocorrelation operation for CWRU bearing data set. (**a**) CWRU bearing data set 169; (**b**) CWRU bearing data set 158.

**Figure 19 entropy-24-00197-f019:**
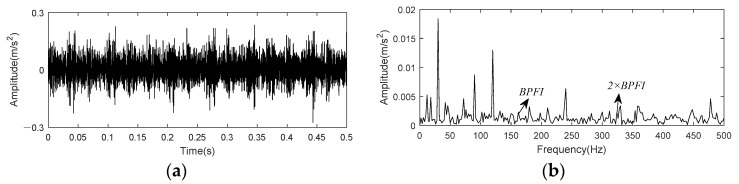
Time domain and envelope spectrum of CWRU bearing data set 169. (**a**)Time domain waveform of raw data; (**b**) Envelope spectrum of raw data.

**Figure 20 entropy-24-00197-f020:**
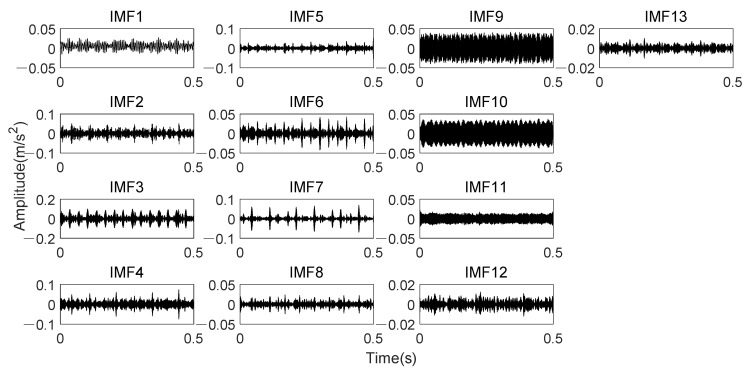
Time domain waveforms of thirteen IMFs obtained by K-adaptive VMD from date set 169.

**Figure 21 entropy-24-00197-f021:**
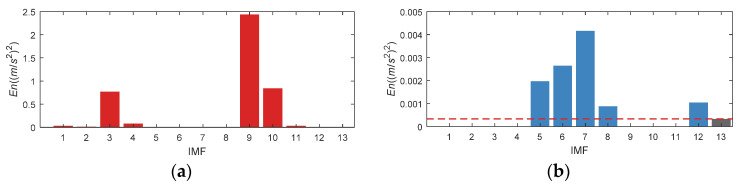
*En* of CWRU bearing data set 169. (**a**) Original *En*; (**b**) Updated *En*.

**Figure 22 entropy-24-00197-f022:**
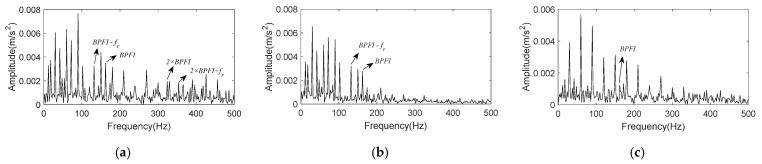
Envelope spectrum of reconstructed signal for CWRU data set 169. (**a**) Reconstruction signal obtained by K-adaptive VMD; (**b**) Reconstruction signal obtained by FBE-VMD; (**c**) Reconstruction signal obtained by TVMD.

**Figure 23 entropy-24-00197-f023:**
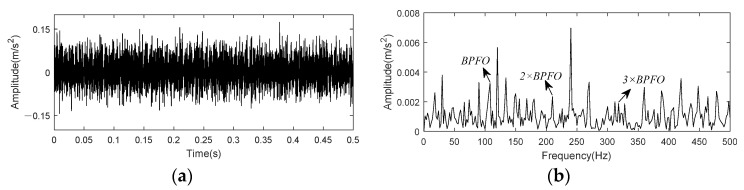
Time domain and envelope spectrum of CWRU bearing data set 158. (**a**)Time domain waveform of raw data; (**b**) Envelope spectrum of raw data.

**Figure 24 entropy-24-00197-f024:**
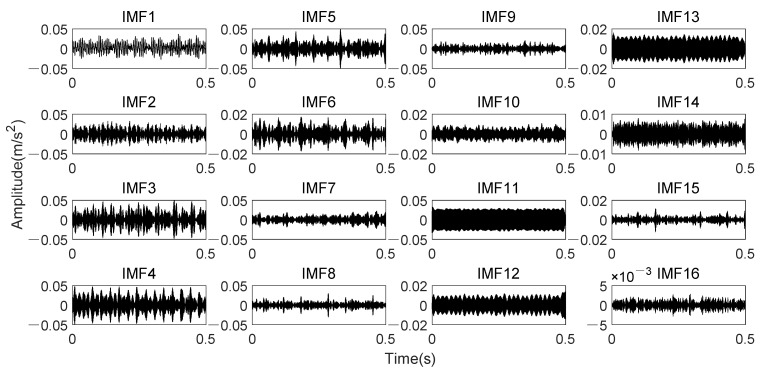
Time domain waveforms of sixteen IMFs obtained by K-adaptive VMD from date set 158.

**Figure 25 entropy-24-00197-f025:**
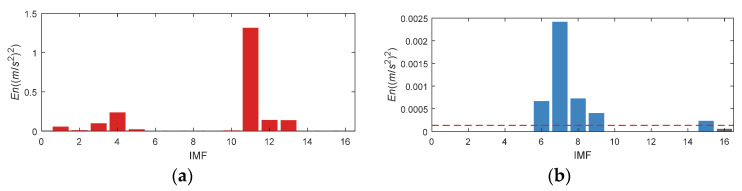
*En* of CWRU bearing data set 158. (**a**) Original *En*; (**b**) Updated *En*.

**Figure 26 entropy-24-00197-f026:**
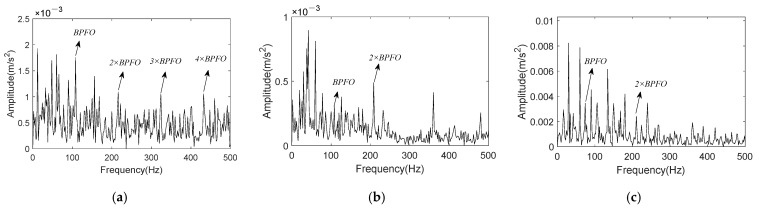
Envelope spectrum of reconstructed signal for CWRU data set 158. (**a**) Reconstruction signal obtained by K-adaptive VMD; (**b**) Reconstruction signal obtained by FBE-VMD; (**c**) Reconstruction signal obtained by TVMD.

**Figure 27 entropy-24-00197-f027:**
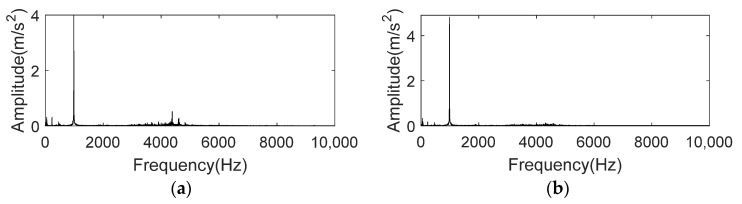
Frequency spectrum of autocorrelation function for IMS bearing data. (**a**) The 5410 min of IMS bearing data; (**b**) The 100 min of IMS bearing data.

**Figure 28 entropy-24-00197-f028:**
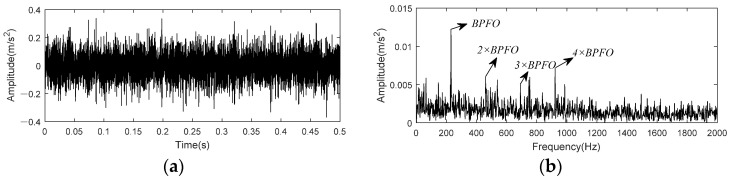
Time domain and envelope spectrum of IMS bearing data. (**a**)Time domain waveform of raw data; (**b**) Envelope spectrum of raw data.

**Figure 29 entropy-24-00197-f029:**
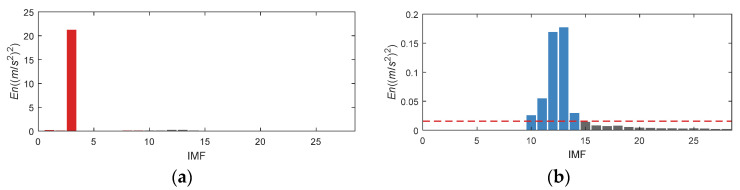
*En* of IMS bearing data. (**a**) Original *En*; (**b**) Updated *En*.

**Figure 30 entropy-24-00197-f030:**
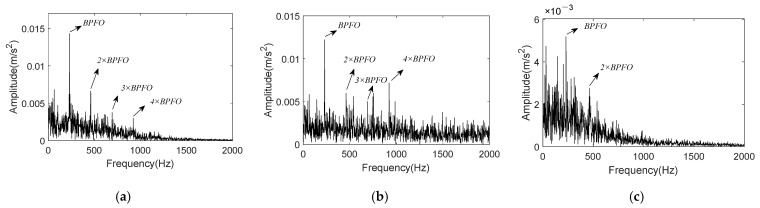
Envelope spectrum of reconstructed signal for the 5410min data of IMS dataset. (**a**) Reconstruction signal obtained by K-adaptive VMD; (**b**) Reconstruction signal obtained by FBE-VMD; (**c**) Reconstruction signal obtained by TVMD.

**Table 1 entropy-24-00197-t001:** Noise reduction results obtained by K-adaptive VMD with different number of samples.

Number of Samples	*K*	Selected IMFs	Center Frequency of the Selected IMFs (Hz)	SNR of Reconstructed Signal (dB)
500	31	1, 2, 6	7.04, 29.19, 101.64	5.00
800	34	1, 2, 5, 6	7.38, 30.19, 98.92, 107.53	4.09
1000	35	1, 2, 6	8.23, 29.42, 100.71	5.69
1200	32	1, 2, 5	7.19, 29.88, 99.01	4.02
1500	32	1, 2, 5	7.10, 29.92, 99.35	3.95
2000	32	1, 2, 5	7.68, 29.53, 99.40	3.77

**Table 2 entropy-24-00197-t002:** Comparison of noise reduction effects of AM-FM signals under different noise level.

Signal	Method	SNR of Reconstructed Signal (dB)	*K*	Selected IMFs	CF of the Selected IMFs (Hz)	Running Time(s)	Complexity
*f_isg_*_1_ (6 dB)	FBE-VMD	−0.29	12	7, 8, 9, 10, 12	244.05, 293.80, 338.51, 377.83, 481.32	——	high
	TVMD	−0.07	3	3	361.3	2.45	low
	K-adaptive VMD	13.44	17	1, 2, 4	7.29, 30.37, 100.13	35.44	middle
*f_sig_*_1_ (−6 dB)	FBE-VMD	−1.21	15	11, 13	344.63, 406.17	——	high
	TVMD	−0.76	3	2	165.57	0.62	low
	K-adaptive VMD	5.69	35	1, 2, 6	8.23, 29.42, 100.71	162.45	middle

**Table 3 entropy-24-00197-t003:** Comparison of noise reduction effects of inner race bearing fault simulation signals under different noise level.

SNR of Original Signal (dB)	Method	*K*	Selected IMFs	SNR of Reconstructed Signal (dB)
−6	FBE-VMD	9	8	−0.82
	TVMD	3	3	4.01
	K-adaptive VMD	37	14, 15, 16, 17	3.38
−13	FBE-VMD	14	6, 8, 9	−5.25
	TVMD	5	3 4	−4.51
	K-adaptive VMD	35	15, 16, 26	−1.44

**Table 4 entropy-24-00197-t004:** Information of CWRU bearing data.

Data Set	State	Fault Diameter	Motor Load	Motor Speed (rpm)	Motor Ratational Frequency *f_r_* (Hz)	Fault Frequencies (Hz)
169	Inner race fault	0.014″	0	1796	29.93	162.09
158	Outer race fault	0.007″	1	1773	29.55	105.94

**Table 5 entropy-24-00197-t005:** CFBs of IMF1~13 for CWRU bearing data set 169 obtained by K-adaptive VMD. The IMFs in bold means that their corresponding *En* should be set to 0.

No	CFB (Hz)	No	CFB (Hz)	No	CFB (Hz)	No	CFB (Hz)
**IMF1**	**[22, 162]**	IMF5	[1920, 2004]	**IMF9**	**[4138, 4146]**	IMF13	[5314, 5350]
**IMF2**	**[444, 528]**	IMF6	[2252, 2304]	**IMF10**	**[4260, 4264]**		
**IMF3**	**[868, 924]**	IMF7	[2692, 2736]	**IMF11**	**[4384, 4384]**		
**IMF4**	**[1380, 1436]**	IMF8	[3102, 3194]	IMF12	[4786, 4862]		

**Table 6 entropy-24-00197-t006:** CFBs of IMF1~16 for CWRU bearing data set 158 obtained by K-adaptive VMD. The IMFs in bold means that their corresponding *En* should be set to 0.

No	CFB (Hz)	No	CFB (Hz)	No	CFB (Hz)	No	CFB (Hz)
**IMF1**	**[162, 166]**	**IMF5**	**[1438, 1442]**	IMF9	[3214, 3258]	**IMF13**	**[4254, 4258]**
**IMF2**	**[644, 688]**	IMF6	[1970, 1998]	**IMF10**	**[3996, 4020]**	**IMF14**	**[4380, 4400]**
**IMF3**	**[868, 936]**	IMF7	[2262, 2290]	**IMF11**	**[4136, 4140]**	IMF15	[4830, 4882]
**IMF4**	**[1084, 1108]**	IMF8	[2762, 2806]	**IMF12**	**[4246, 4258]**	IMF16	[5318, 5338]

**Table 7 entropy-24-00197-t007:** Entropy value and *r* calculated by RBF-FuzzyEn, FuzzyEn1, FuzzyEn2, ApEn, SampEn and MFE.

State of Bearing	N		IF		OF	
	*r*	Entropy	*r*	Entropy	*r*	Entropy
RBF-FuzzyEn	0.0131	1.0523	0.0131	0.5963	0.0131	0.4784
FuzzyEn1	0.0155	0.1336	0.0038	0.1325	0.0025	0.1055
FuzzyEn2	0.0155	1.1785	0.0038	1.8773	0.0025	2.1545
ApEn	0.0155	1.1797	0.0038	1.4369	0.0025	1.6343
SampEn	0.0155	1.1520	0.0038	1.3676	0.0025	1.5938
MFE	0.0116	1.7887	0.0029	0.2171	0.0155	0.2502

**Table 8 entropy-24-00197-t008:** CFBs of IMF1~28 for IMS Bearing data obtained by K-adaptive VMD. The components in bold means that their corresponding *En* should be set to 0.

No	CFB (Hz)	No	CFB (Hz)	No	CFB (Hz)	No	CFB (Hz)
**IMF1**	**[50, 126]**	**IMF8**	**[3208, 3248]**	IMF15	[5076, 5124]	IMF22	[7574, 7630]
**IMF2**	**[464, 528]**	**IMF9**	**[3488, 3532]**	IMF16	[5368, 5432]	IMF23	[7902, 7966]
**IMF3**	**[982, 986]**	IMF10	[3758, 3810]	IMF17	[5690, 5738]	IMF24	[8266, 8338]
**IMF4**	**[1448, 1532]**	IMF11	[4096, 4140]	IMF18	[5970, 6022]	IMF25	[8662, 8746]
**IMF5**	**[1870, 1918]**	IMF12	[4336, 4376]	IMF19	[6296, 6352]	IMF26	[9226, 9286]
**IMF6**	**[2356, 2444]**	IMF13	[4582, 4602]	IMF20	[6554, 6718]	IMF27	[9528, 9580]
**IMF7**	**[2866, 2942]**	IMF14	[4830, 4854]	IMF21	[7152, 7240]	IMF28	[9832, 9896]

## Data Availability

Bearing data set of CWRU at https://engineering.case.edu/bearingdatacenter (accessed on 25 December 2019).

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
