# Peer review of "Feature Enhancement Method of Rolling Bearing Based on K-Adaptive VMD and RBF-Fuzzy Entropy"

_entropy, 2022, doi:10.3390/e24020197_

Round 1
Reviewer 1 Report
In this work, a feature enhancement method is presented. It is applied to rolling bearing fault detection. In general, it is based on k-adaptive VMD and RBF-fuzzy entropy. Although the work is interesting, some issues have to be addressed and clarified.
In literature, there are some works that solve the problem of bearing faults by using VMD. In the first place, nothing is said about it (please review and discuss them)! On the other hand, if such works solve the bearing fault problem, what is the novelty of your work? In a practical sense, is your work only a method to improve efficiency in a solved problem?
Basic theory needs references!
In the introduction section, the selection of a K value without a plenty justification is criticized, but in your algorithm, several values are set without justification (section 3.1.1). Please comment on this! Also, put and describe examples about the impact and results for different values. This is fundamental!
In Figure 3, show the individual signals of equation 14.
Again, the selection of -6 dB is not justified.
In equation 14, the signals have frequencies very separated. What are the results for signals with close frequencies (include an example)? Also, transient frequency signals have to be analyzed. Finally, the impact of the time window length has to be discussed (e.g., show the results of your method for a signal with a different number of samples).
IMFs in Figure 4 are not clear (provide the time domain IMFs).
It is not clear the selection of k=13, 35, 150. Please justify!
The application of the proposed algorithm is not shown. For instance, in line 422, it is said that the K value equal to 35 is found, but the procedure according to the algorithm is not shown. For instance, the evolution of the K value. The number of iterations, the IMFs , the En values, etc. This is fundamental.
In table 1, some methods are compared but their settings are not mentioned. Readers cannot reproduce your results. Also, other qualitative and quantitative features have to be discussed (e.g., complexity, computational burden, time window length, etc.). Your method can provide the best results in terms of error but it can have the maximum computational burden.
Time domain signals for the IMFs of table 4 have to be included (also show their spectra) and compare them!
Other feature extraction methods have to be included and compared. It is not clear how your method is better for feature extraction.
Please improve the quality of the figures.
Author Response
Dear reviewer,
Thank you for your careful review of this paper and your valuable comments on the revision of the paper. In response to your comments, we have made revisions one by one, as detailed below.
Comment 1: In literature, there are some works that solve the problem of bearing faults by using VMD. In the first place, nothing is said about it (please review and discuss them)! On the other hand, if such works solve the bearing fault problem, what is the novelty of your work? In a practical sense, is your work only a method to improve efficiency in a solved problem?
Reply 1: References [9-18] ([10-19] in the revised manuscript) in the manuscript are all improved methods of VMD for solving the problem of bearing faults. We have supplemented the corresponding content in the revised manuscript, explaining that these works are all about bearing fault diagnosis. The novelty of our work is that it does not to set a limit on the range of K, obtains K according to the ADCF phenomenon adaptively, and proposes a new IMF selecting method. In a practical sense, our work can also be used for reference in the fault diagnosis of other rotating machinery.
Comment 2: Basic theory needs references!
Reply 2: We have added the references of basic theory in section 2.
Comment 3: In the introduction section, the selection of a K value without a plenty justification is criticized, but in your algorithm, several values are set without justification (section 3.1.1). Please comment on this! Also, put and describe examples about the impact and results for different values. This is fundamental!
Reply 3: In section 3.1.1, we supplement descriptions of the meaning of VMD parameters, the influence of parameters on VMD and the foundation of parameters determination.
Comment 4: In Figure 3, show the individual signals of equation 14.
Reply 4: We supplement Figure 3 with the time-domain waveforms of the three components in equation 14 (equation 17 in the revised manuscript).
Comment 5: Again, the selection of -6 dB is not justified.
Reply 5: In Section 4.1.1, we explain the reason for adding -6dB to the simulated signal based on the SNR of the simulated signal in references.
Comment 6: In equation 14, the signals have frequencies very separated. What are the results for signals with close frequencies (include an example)? Also, transient frequency signals have to be analyzed. Finally, the impact of the time window length has to be discussed (e.g., show the results of your method for a signal with a different number of samples).
Reply 6: We added a simulated signal composed of close frequencies, so as to verify the noise reduction and decomposition ability of K-adaptive VMD when the main frequencies of the signals are relatively close. This research content is in the revised manuscript section 4.1.2.
We are not sure whether the transient frequency you mentioned in comment 6 is a variable frequency signal. The method proposed in this paper is aimed at the feature enhancement of bearing at a constant speed, so the verification content of transient frequency is not added.
Finally, we verified the noise reduction ability of K-adaptive VMD for signals of different lengths, and the related results are shown in Table 1 of the revised manuscript.
Comment 7: IMFs in Figure 4 are not clear (provide the time domain IMFs).
Reply 7: We have supplemented the time domain waveforms of all IMFs in Figure 4.
Comment 8: It is not clear the selection of k=13, 35, 150. Please justify!
Reply 8: We changed the original K=13, 35 and 150 to K=15, 35 and 150, and explained the choice of these three values in section 4.1.1.
Comment 9: The application of the proposed algorithm is not shown. For instance, in line 422, it is said that the K value equal to 35 is found, but the procedure according to the algorithm is not shown. For instance, the evolution of the K value. The number of iterations, the IMFs, the En values, etc. This is fundamental.
Reply 9: We supplemented center frequency changes of IMFs, the IMFs, etc. in the application of the proposed method, as shown in Figure 12a, 20 and 24.
Comment 10: In table 1, some methods are compared but their settings are not mentioned. Readers cannot reproduce your results. Also, other qualitative and quantitative features have to be discussed (e.g., complexity, computational burden, time window length, etc.). Your method can provide the best results in terms of error but it can have the maximum computational burden.
Reply 10: The parameters of methods FBE-VMD and TVMD are described in section 4.1.1. We have added running time and complexity analysis of the method in Table 1 (Table 2 of the revised manuscript). The relevant analysis of the contents added in the table is also supplemented in the revised manuscript.
Comment 11: Time domain signals for the IMFs of table 4 have to be included (also show their spectra) and compare them!
Reply 11: We supplement the time domain waveforms (Figures 20 and 24 in the revised manuscript) of IMFs and spectra analysis of reconstructed signal corresponding to those IMFs.
Comment 12: Other feature extraction methods have to be included and compared. It is not clear how your method is better for feature extraction.
Reply 12: On the basis of the original three feature extraction methods, we added other three feature extraction methods for analysis and comparison, namely ApEn, SampEn and MFE.
Comment 13: Please improve the quality of the figures.
Reply 13: We have modified Figures 3, 4, 9, 10, 13, 14, 18, 21 and 25(Figure 3, 4, 9, 13, 16, 17, 22, 26 and 30 in the revised manuscript).
Reviewer 2 Report
The manuscript is dedicated to the problem of an analysis of rolling bearing signals. The authors propose to apply the method that applies variational mode decomposition with K determined adaptively and radial based function fuzzy entropy. In my opinion, the method proposed by authors is original, interesting and worth publishing. In general the manuscript is of high scientific quality. My comments and suggestions concerning the manuscript are following:
- In the introduction, authors focus on numerical analysis of the signal of rolling bearings. However, apart from the analysis of vibrations or noise manufacturers of rolling bearings use systems that measure technological parameters describing the quality of rolling bearings. For example, such systems can measure rolling resistance (resistant torque). We can find such information for example in the following work: https://doi.org/10.1016/j.proeng.2017.06.167
- In Fig. 1 authors give the flowchart of the K-adaptive VMD, where the autocorrelation function is applied. There are a number of such functions. Could authors give the equation for the function they apply in their study?
- Fig 4 – please make the legend larger as it is difficult to read.
- Section “Conclusions” should be significantly extended, as it is too short
After the corrections, I suggest I will be pleased to recommend publishing the manuscript.
Author Response
Dear reviewer,
Thank you for your careful review of this paper. In response to your comments, we have made revisions one by one, as detailed below.
Comment 1: In the introduction, authors focus on numerical analysis of the signal of rolling bearings. However, apart from the analysis of vibrations or noise manufacturers of rolling bearings use systems that measure technological parameters describing the quality of rolling bearings. For example, such systems can measure rolling resistance (resistant torque). We can find such information for example in the following work: https://doi.org/10.1016/j.proeng.2017.06.167
Reply 1: We have supplemented the word about measurement of rolling bearing vibration signals in the introduction section.
Comment 2: In Fig. 1 authors give the flowchart of the K-adaptive VMD, where the autocorrelation function is applied. There are a number of such functions. Could authors give the equation for the function they apply in their study?
Reply 2: We supplemented the autocorrelation function used in this paper as equation 12 in the revised manuscript.
Comment 3: Fig 4 – please make the legend larger as it is difficult to read.
Reply 3: We modified Figure 4 so as to easy reading.
Comment 4: Section “Conclusions” should be significantly extended, as it is too short
Reply 4: We enriched the conclusions and specific details are described in revised version.
Round 2
Reviewer 1 Report
All my comments and suggestions have been properly addressed. This Reviewer recommends the manuscript acceptance.